# RETHINKING LoRA FOR PRIVACY-PRESERVING FEDERATED LEARNING IN LARGE MODELS

**Jin Liu [1], Yinbin Miao[1], Ning Xi[1],[*] Junkang Liu[2]**

[1] School of Cyber Engineering, Xidian University

[2] College of Intelligence and Computing, Tianjin University

`{jinliu9787, junkangliukk}@gmail.com, {ybmiao, nxi}@xidian.edu.cn`

## ABSTRACT

Fine-tuning large vision models (LVMs) and large language models (LLMs) under differentially private federated learning (DPFL) is hindered by a fundamental privacy-utility trade-off. Low-Rank Adaptation (LoRA), a promising parameter-efficient fine-tuning (PEFT) method, reduces computational and communication costs by introducing two trainable low-rank matrices while freezing pre-trained weights. However, directly applying LoRA in DPFL settings leads to performance degradation, especially in LVMs. Our analysis reveals three previously underexplored challenges: (1) gradient coupling caused by the simultaneous update of two asymmetric low-rank matrices, (2) compounded noise amplification under differential privacy, and (3) sharpness of the global aggregated model in the parameter space. To address these issues, we propose LA-LoRA (**L**ocal **A**lternating **LoRA**), a novel approach that decouples gradient interactions and aligns update directions across clients to enhance robustness under stringent privacy constraints. Theoretically, LA-LoRA strengthens convergence guarantees in noisy federated environments. Extensive experiments demonstrate that LA-LoRA achieves state-of-the-art (SOTA) performance on Swin Transformer and RoBERTa models, showcasing robustness to DP noise and broad applicability across both LVMs and LLMs. For example, when fine-tuning the Swin-B model on the Tiny-ImageNet dataset under a strict privacy budget ($\epsilon = 1$), LA-LoRA outperforms the best baseline, RoLoRA, by 16.83% in test accuracy. Code is provided in `https://github.com/junkangLiu0/LA-LORA`.

## 1 INTRODUCTION

As foundation models such as GPT (Achiam et al., 2023), BERT (Devlin et al., 2019), and ViT (Dosovitskiy et al., 2020) scale in size and capacity, adapting them to downstream tasks increasingly relies on vast and heterogeneous datasets. However, centralized collection is becoming impractical due to data silos and rising concerns over user privacy (Liu et al., 2024b;a). **Federated learning (FL, McMahan et al. (2017))** offers a privacy-preserving solution by enabling decentralized model training without sharing raw data across clients.

Despite this promise, applying large-scale models in FL remains a major challenge. These models typically consist of billions of parameters, making full-model fine-tuning computationally expensive and communication-heavy. To mitigate this, **parameter-efficient fine-tuning (PEFT)** methods, such as Low-Rank Adaptation (LoRA, Hu et al. (2021)), have been integrated into FL. Approaches like FedIT (Zhang et al., 2024a) freeze the majority of the model weights and only update a lightweight set of LoRA parameters, reducing communication overhead to less than 0.1% of the full model while preserving performance (Xie et al., 2026; Ouyang et al., 2024; Ke et al., 2025; Qiu et al., 2025; Zhou et al., 2025a;b; Wang et al., 2025; Wu et al., 2025; Qi et al., 2025a; Zhao et al., 2026; Bellavia et al., 2024; Sunmola et al., 2025; Yan et al., 2025; Edstedt et al., 2024; Meng et al., 2025; Qi et al., 2025b; Feng et al., 2026; 2024a; Zhou et al., 2025c).

**However, privacy remains a critical concern in FL**. Although raw data is not directly shared, transmitting gradients or model updates can still expose sensitive information. Prior work has shown

---
[*]Corresponding author.

Figure 1: The illustration of DP-LoRA, FFA-LoRA, RoLoRA, and LA-LoRA. DP-LoRA updates both noisy $A$ and $B$ simultaneously and sends them to the server for aggregation. FFA-LoRA freezes $A$, updates only the noisy $B$, and sends it to the server. RoLoRA alternately updates noisy $A$ and $B$ across rounds. Our LA-LoRA alternately updates noisy $A$ and $B$ within each local round.

that adversaries may reconstruct private data from such gradients (Mu et al., 2024b;a; Xiao et al., 2024). To provide formal privacy guarantees, **differential privacy (DP, Dwork (2006))** is commonly integrated into federated optimization.

While differentially private federated learning (DPFL) with PEFT methods (e.g., DP-LoRA (Liu et al., 2025e)) provides privacy guarantees, it often incurs reduced performance due to federated constraints. Federated training introduces additional complexities, including non-iid data distributions, noisy updates, and difficulties in aggregating LoRA modules. In practice, LoRA-based fine-tuning under privacy constraints often suffers from severe performance degradation (Feng et al., 2024b; Liu et al., 2024b;a; 2025c;a;b;d; 2026). Beyond the prior finding of DP noise amplification in FFA-LoRA, we identify two further challenges, leading to three key issues of LoRA in DPFL :

- **Gradient coupling.** Simultaneous updates to LoRA's asymmetric matrices cause gradient interference, destabilizing training, especially under DP noise and non-iid distributions.
- **Amplified DP noise.** LoRA's semi-quadratic update structure amplifies the impact of injected DP noise, severely degrading learning stability.
- **Sharp global solutions.** LoRA's low-rank constraints reduce client capacity, often resulting in sharp global minima after aggregation, compromising robustness and generalization.

To overcome these, we propose **LA-LoRA**, a novel framework built on a structural rethinking of how LoRA fits into DPFL. Its core is a **local alternating update mechanism** that decouples gradient interference and suppresses DP noise. To further enhance stability and generalization, LA-LoRA incorporates a simple yet effective **low-pass smoothing filter**. We provide theoretical evidence that our update scheme ensures stable optimization while preserving the low-rank structure of LoRA. Extensive experiments on both **image classification** and **language understanding** tasks validate the effectiveness of LA-LoRA. Results show that LA-LoRA achieves strong performance while offering privacy protection and optimization stability. Our contributions are summarized as follows:

- We identify two overlooked challenges of applying LoRA in DPFL: gradient coupling and aggregation sharpness. Along with noise amplification, these motivate the design of our algorithm.
- We propose LA-LoRA, a novel algorithm that alternates local updates of LoRA components to mitigate gradient interference, noise amplification, and aggregation instability. A low-pass smoothing filter further enhances cross-client consistency and stability.
- Theoretically, we prove that alternating updates yield unique closed-form solutions and ensure stable low-rank optimization, mitigating projection distortion in federated settings.
- We validate our algorithm on both vision (Swin Transformer) and language (RoBERTa) models, covering image classification and NLP tasks. LA-LoRA outperforms SOTA privacy-preserving federated LoRA methods on various tasks and demonstrates significant performance gains.

## 2 BACKGROUND AND RELATED WORK

**Definition 1** (($\epsilon, \delta$)-DP)**.** *Let $\epsilon > 0$ and $0 < \delta < 1$. A random mechanism $\mathcal{M} : \mathcal{X}^n \to \mathcal{O}$ satisfies ($\epsilon, \delta$)-DP if, for any two neighboring datasets $\mathcal{D}$ and $\mathcal{D}'$ differing in one sample, and for*

*any measurable subset $\mathcal{U} \subseteq \mathcal{O}$ of possible outputs,*

$$\Pr[\mathcal{M}(\mathcal{D}) \in \mathcal{U}] \leq e^\epsilon \cdot \Pr[\mathcal{M}(\mathcal{D}') \in \mathcal{U}] + \delta. \qquad (1)$$

**Differentially private federated learning (DPFL).** A general FL system with a central server and $N$ clients. Each client $i$ holds a local dataset $\mathcal{D}_i$ and performs local training on it. The server aggregates the model updates from all clients and updates the global model $W$. Our privacy goal is sample-level protection (changes to any single data sample should not significantly affect the output). Building on Noble et al. (2022), we provide $(\epsilon, \delta)$-DP guarantees. We perform privacy accounting via **Rényi DP (RDP, Mironov (2017))**, composing per-step privacy loss for the subsampled Gaussian (Wang et al., 2019) with per-sample clipping, and finally convert to $(\epsilon, \delta)$-DP.

To achieve DP in FL, each client $i$ clips the per-sample gradients to a fixed $\ell_2$ norm to bound sensitivity, then adds Gaussian noise to the aggregated clipped gradients locally, protecting the contribution of individual training examples. For a local mini-batch $\mathcal{B}_i$, the privatized update is computed as:

$$g_{ij} = \nabla\mathcal{L}_{ij}/\max\left(1, \|\nabla\mathcal{L}_{ij}\|_2/C\right), \forall j \in \mathcal{B}_i, \quad g_i = \left(\sum\nolimits_{j \in \mathcal{B}_i} g_{ij} + \mathcal{N}(0, C^2\sigma^2)\right)/|\mathcal{B}_i|, \quad (2)$$

where $\mathcal{L}_{ij}$ denotes the loss for sample $j$ on client $i$, $C$ is the clipping norm bound, and $\sigma$ is the Gaussian noise multiplier determined by the privacy budget $(\epsilon, \delta)$.

**Parameter-efficient fine-tuning (PEFT).** The rapid scaling of modern pre-trained models has led to an increased demand for fine-tuning in resource-constrained environments. PEFT tackles this by training a small set of parameters while freezing most backbone weights, with representative methods including adapters (Houlsby et al., 2019), prefix-tuning (Li & Liang, 2021), prompt-tuning (Lester et al., 2021), LoRA (Hu et al., 2021; Tian et al., 2024) and other methods (Shin et al., 2023; Zheng et al., 2026). LoRA is particularly popular for its simplicity and strong performance under minimal parameter overhead: it augments a pre-trained weight matrix $W_0 \in \mathbb{R}^{m \times n}$ with two low-rank matrices, an up-projection $B \in \mathbb{R}^{m \times r}$ and a down-projection $A \in \mathbb{R}^{r \times n}$ with $r \ll \min\{m, n\}$, and reparameterizes the weights as $W = W_0 + sBA$ while keeping $W_0$ fixed. $B$ is initialized to a zero matrix to suppress early updates, while $A$ uses a random Gaussian initialization.

**PEFT in FL.** PEFT in FL has been explored to reduce communication overhead, computational cost, and privacy risks. Accordingly, various strategies have been proposed, such as adapter-based (Cai et al., 2023; Ghiasvand et al., 2024), prompt-based (Zhao et al., 2023; Qiu et al., 2023), and selective tuning approaches (Yu et al., 2023). Recently, LoRA-based methods have received increasing attention in FL. FedIT (Zhang et al., 2024a) directly incorporates LoRA into the standard FedAvg framework for instruction tuning. RoLoRA (Chen et al., 2025) addresses heterogeneity by alternately optimizing $A$ and $B$ across communication rounds. FedSA-LoRA (Guo et al., 2025) uploads only $A$ to the server for aggregation, keeping $B$ local to support personalized adaptation. Other works explore heterogeneous configurations (Cho et al., 2024; Wang et al., 2024) or personalized decomposition (Qi et al., 2024) to improve adaptability under system and data heterogeneity.

**LoRA in DPFL.** In DPFL, LoRA-based methods typically apply DP mechanisms to the low-rank matrices $A$ and $B$, rather than the full model parameters, which significantly reduces computational and communication overhead. DP-LoRA (Liu et al., 2025e) computes per-sample gradients for $A$ and $B$ locally, clips them, adds Gaussian noise, and sends privatized updates for aggregation. FFA-LoRA (Sun et al., 2024) freezes the down-projection matrix $A$ and trains only the up-projection $B$, injecting calibrated noise into its updates. This design avoids noise amplification, but at the cost of reduced

Table 1: Comparison of LoRA-based methods in DPFL. "✓" denotes support, "✗" denotes not considered. "**Exp.**" indicates the effective expression ability under DP.

| Method | DP | LVMs | Exp. | Speed |
|---|---|---|---|---|
| DP-LoRA | ✓ | ✗ | mid | slow |
| FFA-LoRA | ✓ | ✗ | low | slow |
| RoLoRA | ✗ | ✗ | mid | mid |
| LA-LoRA | ✓ | ✓ | high | fast |

model expressiveness. Despite recent efforts, differentially private federated LoRA methods still suffer from noticeable performance degradation and remain underexplored. Figure 1 shows the differences in the ideas of existing methods, while Table 1 compares their actual characteristics.

## 3 LIMITATIONS OF LORA UNDER PRIVACY-PRESERVING FL

Although LoRA is widely used for efficient adaptation, its behavior under differential privacy is still poorly understood. We show that DP interacts with the low-rank parameterization and causes failure

modes that are further amplified in federated learning due to data heterogeneity and noisy local updates. We therefore focus on DPFL and include centralized DP experiments in Appendix B.6.

## 3.1 GRADIENT COUPLING IN ASYMMETRIC UPDATES

We find that simultaneously updating the two LoRA matrices $A$ and $B$ leads to intrinsic gradient coupling that destabilizes training under DP noise and data heterogeneity. In LoRA, $A \in \mathbb{R}^{r \times n}$ projects the input into a lower-dimensional subspace, while $B \in \mathbb{R}^{m \times r}$ maps this low-rank representation back to the original output space. Their asymmetric roles give rise to distinct gradient behaviors. More precisely, the gradients of the loss $\mathcal{L}$ with respect to $A$ and $B$ are interdependent:

$$\nabla_A \mathcal{L} = sB^\top(\nabla_W \mathcal{L}), \quad \nabla_B \mathcal{L} = s(\nabla_W \mathcal{L})A^\top. \tag{3}$$

Thus, the update direction for $A$ (resp. $B$) is re-parameterized by the current $B$ (resp. $A$). When both matrices are updated in the same step, the latent basis defined by $A$ can shift while $B$ adapts to outdated directions, creating *representation drift* and a mismatch between the two gradient pathways. In the presence of DP perturbations and non-iid data, this coupling makes the trajectory more sensitive to perturbations and client drift, leading to oscillations and unstable convergence.

To quantify this coupling, we compute the cosine similarity between $\nabla_A \mathcal{L}$ and $\nabla_B \mathcal{L}$ during each training step under the experimental setup described in Section 6. As shown in Figure 2(a), DP-LoRA (simultaneous updates) maintains persistently lower similarity, reflecting strong coupling and mismatch. LA-LoRA(-filter) (local alternating updates) achieves much higher similarity. This alignment difference manifests in training dynamics and final performance: Figure 2 (b) shows that the simultaneous update of $A$ and $B$ in DP-LoRA leads to a higher test loss, whereas LA-LoRA(-filter) achieves a smoother convergence. Figure 2(c) further indicates that this coupling harms test accuracy. We observe similar trends in a non-private federated setup (Appendix B.7), which suggests that gradient alignment is inherently beneficial, while DP noise further amplifies the coupling issue.

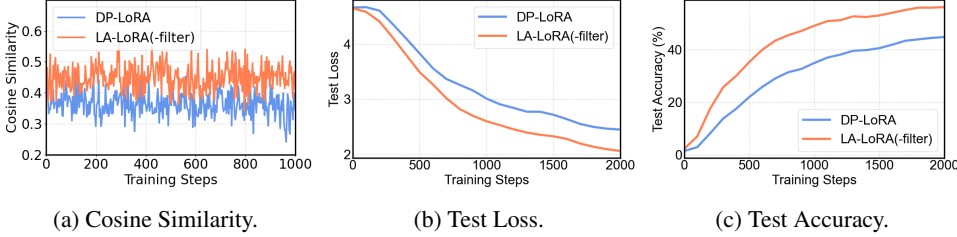

(a) Cosine Similarity.  (b) Test Loss.  (c) Test Accuracy.

Figure 2: Comparison of cosine similarity between $\nabla_A \mathcal{L}$ and $\nabla_B \mathcal{L}$, test loss and test accuracy for Swin-T on CIFAR-100 ($\epsilon = 3$). LA-LoRA(-filter) uses local alternating updates without smoothing.

## 3.2 STRUCTURAL AMPLIFICATION OF DP NOISE

Differential privacy injects noise into local sample gradients to protect client data. Noise is added independently to $A$ and $B$. We ignore the LoRA scaling factor $s$. For client $i$, the resulting noisy low-rank term can be written as:

$$(B_i + \mathcal{N}_{B_i})(A_i + \mathcal{N}_{A_i}) = B_i A_i + B_i \mathcal{N}_{A_i} + \mathcal{N}_{B_i} A_i + \mathcal{N}_{B_i} \mathcal{N}_{A_i}, \tag{4}$$

where $\mathcal{N}_{A_i}$ and $\mathcal{N}_{B_i}$ are Gaussian noises independently sampled per client $i$. This decomposition reveals that the update $B_i A_i$ is perturbed by three terms: $B_i \mathcal{N}_{A_i}$, $\mathcal{N}_{B_i} A_i$ and $\mathcal{N}_{B_i} \mathcal{N}_{A_i}$. The first two are linear in the noise, while $\mathcal{N}_{B_i} \mathcal{N}_{A_i}$ is non-Gaussian and introduces quadratic amplification effect. This structural cascade increases the variance of the aggregated updates and hinders convergence.

We consider a synthetic example. In our setting, $W \in \mathbb{R}^{1024 \times 1024}$ with the dataset QNLI and other configurations described in Section 6. As the Gaussian noise scale $\sigma$ changes, we report the Frobenius norms of the induced perturbations. Figure 3 indicates that for small $\sigma$, LoRA ($\|\mathcal{N}_B \mathcal{N}_A + B\mathcal{N}_A + \mathcal{N}_B A\|_F$) incurs smaller perturbations than full-model ($\|\mathcal{N}_W\|_F$) updates. As $\sigma$ increases, the multiplicative term ($\|\mathcal{N}_B \mathcal{N}_A\|_F$) grows quadratically and becomes the leading contribution, causing the total LoRA perturbation to exceed the full-model curve.

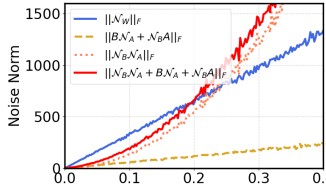

Figure 3: Scaling of perturbation Frobenius norms with $\sigma$ on QNLI.

### 3.3 SHARPNESS IN GLOBAL AGGREGATION

A third distinct challenge arises after global aggregation. It is well established that the geometry of the loss landscape critically affects generalization. Convergence to flat minima, regions with low curvature, has been shown to promote robustness and better generalization, while sharp minima correlate with overfitting and instability (Hochreiter & Schmidhuber, 1997; Kaddour et al., 2022).

We observe that this issue is aggravated in LoRA-based DPFL. In parameter space, each client contributes low-rank factors $A_i$ and $B_i$, and the server aggregates them under naive FedAvg. Unlike full-parameter updates, low-rank factor aggregation alters the geometry of the global update in parameter space. Specifically, heterogeneous factor directions across clients do not align, and their composition produces global updates that consistently land in narrower, high-curvature regions of the landscape. The problem is exacerbated by DP noise, which injects stochastic perturbations that destabilize the already misaligned updates.

Figure 4 visualizes the loss landscapes of Swin-T trained with DP-LoRA and LA-LoRA(-filter) under $\epsilon = 1$. DP-LoRA produces a sharper and more irregular landscape, while LA-LoRA(-filter) yields a flatter and smoother basin. This suggests that simultaneous updates in DPFL may hinder generalization by increasing the curvature of the aggregated model. Table 2 further confirms this sharpness issue via Hessian eigenvalue analysis. See Section 6 for experimental setup.

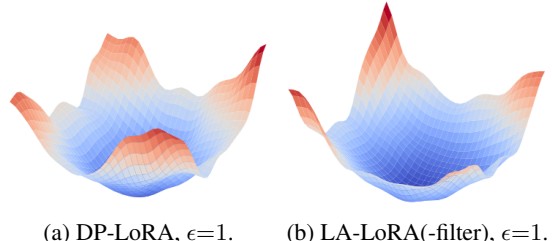

(a) DP-LoRA, $\epsilon$=1.  (b) LA-LoRA(-filter), $\epsilon$=1.

Figure 4: Comparison of global loss landscapes for fine-tuning Swin-T model on CIFAR-100.

## 4 OUR METHOD

We address the aforementioned challenges from two complementary perspectives. At the **optimization level**, we break the tight dependency between the two low-rank factors so that gradients are decoupled, cross-noise terms are avoided, and the update trajectory remains smoother. At the **pre-aggregation level**, we further suppress residual variance by filtering out high-frequency components of DP perturbations on each client before aggregation, effectively smoothing noise and improving generalization. Building on these ideas, we develop LA-LoRA (**L**ocal **A**lternating **Lo**w-**R**ank **A**daptation), illustrated in Figure 5. It combines (i) a **local alternating update strategy** for the first perspective, and (ii) an **optional Gaussian low-pass filter** for the second, jointly improving stability and consistency under DPFL.

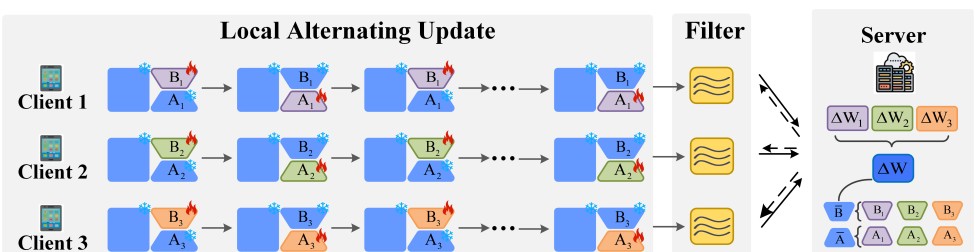

Figure 5: Our LA-LoRA framework.

### 4.1 LOCAL ALTERNATING UPDATE STRATEGY

Instead of simultaneously updating both $A$ and $B$, LA-LoRA adopts an alternating scheme, where the two low-rank matrices are updated in turn within each local training round. For client $i$ at local step $k$ of communication round $t$, we keep one factor fixed and update the other:

- Update $B_i$ if $k$ is odd, keeping $A_i$ fixed: $B_{i,k+1}^t = B_{i,k}^t - \eta_B \nabla_B \mathcal{L}_i(W_0 + s B_{i,k}^t A_{i,k}^t)$.
- Update $A_i$ if $k$ is even, keeping $B_i$ fixed: $A_{i,k+1}^t = A_{i,k}^t - \eta_A \nabla_A \mathcal{L}_i(W_0 + s B_{i,k}^t A_{i,k}^t)$.

Here, $W_0$ denotes the frozen backbone from the server. $\mathcal{L}_i(\cdot)$ is the local training loss on client $i$'s private dataset $\mathcal{D}_i$, $\eta_A$, $\eta_B$ are learning rates, and $s$ is the LoRA scaling. Gradients $\nabla_A \mathcal{L}_i$ and $\nabla_B \mathcal{L}_i$ are taken with respect to the corresponding low-rank matrices.

This local alternating design addresses the three challenges outlined in Section 3:

**Challenge 1 (Gradient coupling).** By updating one matrix at a time, the direct interaction between the two LoRA factors is reduced, which alleviates the tightly coupled dynamics described in Eq. (3):

$$\nabla_B \mathcal{L}_i = s(\nabla_W \mathcal{L}_i) A^\top, \quad \nabla_A \mathcal{L}_i = sB^\top (\nabla_W \mathcal{L}_i). \tag{5}$$

**Challenge 2 (Amplified DP noise).** Since only one matrix is updated and privatized at each step, the same-step multiplicative noise term $\mathcal{N}_{B_i} \mathcal{N}_{A_i}$ in Eq. (4) does not arise. The perturbed update reduces to:

$$\begin{cases} (B_i + \mathcal{N}_{B_i}) A_i = B_i A_i + \mathcal{N}_{B_i} A_i, & \text{if } A_i \text{ is fixed,} \\ B_i(A_i + \mathcal{N}_{A_i}) = B_i A_i + B_i \mathcal{N}_{A_i}, & \text{if } B_i \text{ is fixed.} \end{cases} \tag{6}$$

**Challenge 3 (Sharp global solutions).** Local alternating updates constrain each step to a structured lower-dimensional subspace, the column space of $A_i$ or the row space of $B_i$. This implicit regularization suppresses sensitivity to stochastic noise and client heterogeneity, yielding flatter and more stable global solutions. We report the maximum Hessian eigenvalue, which characterizes the steepest curvature of the loss surface (Sagun et al., 2018; Grosse & Martens, 2016). As shown in Table 2, LA-LoRA consistently achieves smaller eigenvalues than DP-LoRA across datasets and privacy settings, indicating a flatter loss landscape and more stable training dynamics.

## 4.2 SMOOTHING WITH A LOW-PASS FILTER

To further improve training stability under DP, LA-LoRA introduces an optional low-pass smoothing filter applied to the LoRA gradients before aggregation. DP noise often manifests as high-frequency perturbations, which destabilize local updates and amplify sharpness in global aggregation (Zhang et al., 2024b). Our goal is to attenuate these high-frequency components via a lightweight operation that leaves the model architecture and the privacy mechanism unchanged.

Table 2: Maximum Hessian eigenvalue of Swin-B on CIFAR-100 and Tiny-ImageNet.

| Method | CIFAR-100 | Tiny-ImageNet |
|---|---|---|
| DP-LoRA $\epsilon = \infty$ | 42.45 | 44.80 |
| LA-LoRA $\epsilon = \infty$ | 30.12 | 33.25 |
| DP-LoRA $\epsilon = 1$ | 101.62 | 115.36 |
| LA-LoRA $\epsilon = 1$ | 64.77 | 69.53 |

Specifically, we adopt a fixed 1D Gaussian kernel $G_s = \frac{1}{16}[1, 4, 6, 4, 1]$, i.e., the standard 5-tap binomial low-pass filter. For $A \in \mathbb{R}^{r \times n}$, smoothing is applied row-wise along the input feature dimension; for $B \in \mathbb{R}^{m \times r}$, smoothing is applied column-wise along the output dimension. The filter acts along meaningful input/output feature axes while keeping different low-rank components decoupled. Denoting by $*$ the 1D convolution with symmetric padding, the filtered gradients are

$$\widehat{\nabla}_A \mathcal{L}_i[j, :] = G_s * \nabla_A \mathcal{L}_i[j, :], \forall j \in [1, r], \quad \widehat{\nabla}_B \mathcal{L}_i[:, j] = G_s * \nabla_B \mathcal{L}_i[:, j], \forall j \in [1, r], \tag{7}$$

From an optimization perspective, filtering noisy gradients with $G_s$ can be interpreted as approximately imposing a one-dimensional smoothness regularizer along the filtered dimension, discouraging abrupt changes between neighboring entries and inducing a low-pass effect on the update.

In practice, the Gaussian kernel $G_s$ reduces DP-induced fluctuations while preserving the structural semantics encoded in $A$ and $B$. The operation stabilizes local updates and alleviates sharpness in global aggregation with negligible overhead, making it suitable for federated environments.

## 4.3 LA-LoRA FRAMEWORK

**Local client update.** At global round $t \in [T]$, each selected client $i$ fine-tunes only the LoRA factors $(A, B)$ on top of a frozen backbone $W_0$ via the reparameterization $W_0 + sBA$. The factors are initialized as $A_{i,1}^t \leftarrow A^{t-1}$ and $B_{i,1}^t \leftarrow B^{t-1}$. The client performs $K$ local steps. At each step $k \in [K]$, it samples a mini-batch $\mathcal{B}_i \subset \mathcal{D}_i$ of size $\lfloor bR \rfloor$. $b$ is the local data sampling rate, $R$ is the size of $\mathcal{D}_i$. The client performs alternating updates of $B$ and $A$ across local steps.

**Odd steps (update $B$).** For each example $j \in \mathcal{B}_i$, compute the per-example gradient

$$g_{ij} \;=\; \nabla_B \mathcal{L}_i\big(W_0 + sB_{i,k}^t A_{i,k}^t, \, d_j^i\big), \tag{8}$$

Perform per-example $\ell_2$-norm clipping with threshold $C$: $g_{ij} \leftarrow g_{ij}/\max\{1, \|g_{ij}\|_2/C\}$. Aggregate over the mini-batch and inject Gaussian noise to ensure DP:

$$g_i \;=\; 1/(bR)\sum\nolimits_{j\in\mathcal{B}_i} g_{ij} \;+\; C/(bR)\cdot\mathcal{N}(0,\sigma^2). \tag{9}$$

Smooth the noisy gradient with a fixed operator $G_s$ (low-pass filter) to obtain $\widehat{g}_i = G_s * g_i$, then

$$B_{i,k+1}^t \;=\; B_{i,k}^t - \eta_B\,\widehat{g}_i, \qquad A_{i,k+1}^t \;=\; A_{i,k}^t.$$

**Even steps (update $A$).** Repeat the same procedure for $A$ with learning rate $\eta_A$.

After $K$ alternating steps, the client returns the locally updated (and smoothed) factors $\big(A_i^t, B_i^t\big)$ for server-side aggregation. Throughout, $W_0$ remains frozen and only the low-rank adaptation $BA$ is modified; gradient clipping with Gaussian noise provides privacy, while $G_s$ suppresses high-frequency perturbations and stabilizes optimization.

**Server aggregation.** The server averages client uploads:

$$A^t = 1/|\mathcal{C}_t|\sum\nolimits_{i\in\mathcal{C}_t} A_i^t, \quad B^t = 1/|\mathcal{C}_t|\sum\nolimits_{i\in\mathcal{C}_t} B_i^t, \tag{10}$$

and then updates the global model as $W^t = W_0 + sB^t A^t$. After $T$ rounds, the final model is $W^T = W_0 + sB^T A^T$. Here, $\mathcal{C}_t$ is the set of participating clients at round $t$ with $|\mathcal{C}_t| = \lfloor qN \rfloor$, $q$ is the client sampling rate and $N$ is the total number of clients.

## 5 THEORETICAL ANALYSIS

In the following, we state the necessary theorems. Full theoretical details appear in Appendix F.

**Theorem 1** (Privacy guarantee). *Following the privacy analysis in Noble et al. (2022), LA-LoRA ensures that after $T$ communication rounds with $K$ local steps per client, the weight matrix $W^T$ satisfies $(\epsilon, \delta)$-DP for any third party:*

$$\epsilon = \mathcal{O}\left(b\sqrt{TK\log(2/\delta)\log(2T/\delta)}/\sigma\right). \tag{11}$$

*With respect to the server, after $T$ rounds, the accumulated privacy budget satisfies $(\epsilon_s, \delta_s)$-DP,*

$$\epsilon_s = \epsilon\sqrt{N/q}, \quad \delta_s = \delta/2\left(1/q + 1\right). \tag{12}$$

Noise is added to the clipped sample gradients before forming the client update, and the Gaussian low-pass filter is a deterministic function of these noisy updates. By the post-processing invariance of DP (Dwork et al., 2014), this filtering step does not weaken the guarantees in Theorem 1.

**Theorem 2** (Closed-form projected gradients). *Let $B_k \in \mathbb{R}^{m\times r}$ and $A_k \in \mathbb{R}^{r\times n}$ be full rank, i.e., $rank(B_k) = rank(A_k) = r$, and let $s = \alpha/r > 0$ denote the LoRA scaling. In LoRA, updates to $B_k$ and $A_k$ can be obtained by projecting the full gradient onto the column space of $A_k$ and the row space of $B_k$, respectively. Within iteration $k$, we update $B$ first and then $A$. This projection is formulated as a least-squares problem, whose unique solution yields:*

$$\tilde{\nabla}_{B_k}\mathcal{L} \;=\; \frac{1}{s^2}\nabla_{B_k}\mathcal{L}\,(A_k A_k^\top)^{-1}, \qquad \tilde{\nabla}_{A_k}\mathcal{L} \;=\; \frac{1}{s^2}\,(B_{k+1}^\top B_{k+1})^{-1}\,\nabla_{A_k}\mathcal{L}, \tag{13}$$

*where $\nabla_{B_k}\mathcal{L}$, $\nabla_{A_k}\mathcal{L}$ are the gradients defined in Eq. (3). The projected gradients $\tilde{\nabla}_{B_k}\mathcal{L}$, $\tilde{\nabla}_{A_k}\mathcal{L}$ are obtained by solving the least-squares problem.*

Under the full-rank assumption, Theorem 2 yields closed-form projected gradients that use only the local parameter gradients $\nabla_{A_k}\mathcal{L}$ or $\nabla_{B_k}\mathcal{L}$ plus an $r \times r$ solve. This avoids the full model gradient $\nabla_W\mathcal{L}$. In practice, the computation reduces to forming the small Gram matrices $A_k A_k^\top$ or $B_{k+1}^\top B_{k+1}$ and solving a small $r \times r$ system, which is lightweight when $r \ll \min\{m,n\}$.

**Theorem 3** (Stable feature learning). *Assume that, for the input $x$, $BAx$ has dimension $\mathcal{O}(n)$. In LA-LoRA, if we use the learning rate $\eta = \mathcal{O}(1)$ to update $B$ and $A$, it achieves stable feature learning. Moreover, the model update achieves stable feature learning as well with*

$$W_{k+1} = W_k - \eta(\nabla_{W_k}\mathcal{L})Proj_{r(A_k)} - \eta Proj_{c(B_{k+1})}(\nabla_{W_{k+\frac{1}{2}}}\mathcal{L}). \tag{14}$$

*where $Proj_{r(A_k)}$ denotes the orthogonal projection onto the* row *space of $A_k$, and $Proj_{c(B_{k+1})}$ denotes the orthogonal projection onto the* column *space of $B_{k+1}$. Besides, $\eta(\nabla_{W_k}\mathcal{L})Proj_{r(A_k)}, \eta Proj_{c(B_{k+1})}(\nabla_{W_{k+\frac{1}{2}}}\mathcal{L}) \in \mathcal{O}(1)$. However, when doing joint update, the update will introduce additional cross term*

$$\eta^2(B_k^\top B_k)^{-1}B_k^\top(\nabla_{W_k}\mathcal{L})(\nabla_{W_k}\mathcal{L})A_k^\top(A_k A_k^\top)^{-1} \in \mathcal{O}(1).$$

*The across term is indeed the second order term w.r.t $\eta$, but it is same magnitude as $\eta Proj_{c(B_k)}(\nabla_{W_k}\mathcal{L})$ and $\eta(\nabla_{W_k}\mathcal{L})Proj_{r(A_k)}$ in infinite-width NN setting.*

In Theorem 3, our method achieves stable feature learning. Moreover, as the joint update would introduce the cross term with an unignorable magnitude (especially $\eta$ is $\mathcal{O}(1)$ instead of $\mathcal{O}(1/n)$), simultaneous update with scaled gradient descent breaks the clean interpretation of projecting the full gradient onto low-rank subspaces and degrades the performance as our experiment studies show later.

**Theorem 4** (Convergence rate). *Assume for any $i \in [P]$ the matrix $C_i = D_i X$ satisfies the rank-$r$-RIP with constant $\delta_r$ (Assumption 1) and $0 \le \eta \le \frac{1}{1+\delta_r+\frac{1}{P}}$, then LA-LoRA without momentum solves the over-parameterized problem leads to*

$$\mathcal{L}_c(\boldsymbol{B}_{k+1}, \boldsymbol{A}_{k+1}) \le (1-\eta_c)^2 \mathcal{L}_c(\boldsymbol{B}_k, \boldsymbol{A}_k), \tag{15}$$

$$\left\|\sum_i^P B_k^i A_k^i - X_\star\right\|_F^2 \le \frac{1+\delta_r}{1-\delta_r}(1-\eta_c)^{2k}\left\|\sum_i^P B_0^i A_0^i - X_\star\right\|_F^2, \tag{16}$$

*where $\eta_c = 2P(1-\delta_r)\left(\eta - \frac{\eta^2(1+\delta_r+\frac{1}{P})}{2}\right)$.*

**Explanation of Theorem 4.** If each $C_i = D_i X$ satisfies the rank-$r$ RIP with constant $\delta_r$ and the step size obeys $0 \le \eta \le (1+\delta_r+\frac{1}{P})^{-1}$, then momentum-free LA-LoRA is contractive: the objective decreases geometrically as $\mathcal{L}_c(\mathbf{B}_{k+1}, \mathbf{A}_{k+1}) \le (1-\eta_c)^2 \mathcal{L}_c(\mathbf{B}_k, \mathbf{A}_k)$, with $\eta_c = 2P(1-\delta_r)\left(\eta - \frac{\eta^2(1+\delta_r+\frac{1}{P})}{2}\right)$. Consequently, the reconstruction $\widehat{X}_k = \sum_{i=1}^P B_k^i A_k^i$ converges linearly to $X_\star$ in Frobenius norm: $\|\widehat{X}_k - X_\star\|_F^2 \le \frac{1+\delta_r}{1-\delta_r}(1-\eta_c)^{2k}\|\widehat{X}_0 - X_\star\|_F^2$, where $\frac{1+\delta_r}{1-\delta_r}$ reflects the RIP conditioning (smaller $\delta_r$ gives a tighter bound).

## 6 Experiments

We evaluate LA-LoRA on both vision and language tasks to assess its effectiveness and privacy preservation. All experiments are performed on NVIDIA A6000 GPUs.

### 6.1 Experimental setups

**Datasets**. For *image classification*, we use CIFAR-100 and Tiny-ImageNet. For *language under-standing*, we evaluate on four GLUE benchmarks: SST-2, QNLI, QQP, and MNLI.

**Models**. In vision tasks, we employ Swin Transformer backbones (Swin-T and Swin-B), initialized from ImageNet-22K pre-trained weights, which are well-suited to federated environments due to their strong generalization. For language tasks, we use RoBERTa-Base as the backbone model.

**Baselines**. We compare LA-LoRA with three SOTA approaches: **DP-LoRA**, a direct application of LoRA under DPFL constraints. **FFA-LoRA**, which freezes $A$ while updating $B$. **RoLoRA**, which alternates the upload of $A$ and $B$ in communication rounds.

**Hyperparameter settings**. *Image classification*: Federated setup with $N = 8$ clients (sampling rate $q = 0.5$) and default non-iid Dirichlet $\beta = 0.1$. Training runs for $T = 100$ rounds, each selected

client performing $K = 20$ with batch size $\mathcal{B} = 16$. We use LoRA fine-tuning with rank $r = 16$, scaling factor $\alpha = 16$, updating both adapter and classification head. Optimization uses SGD with learning rate decay $\lambda = 0.99$. The learning rate $\eta$ is selected from $\{1e-2, 2e-2, 1e-1, 2e-1\}$. The privacy budget is fixed to $\epsilon \in \{3, 2, 1\}$, with $\delta = 1e-5$ and noise smoothing $\sigma_s = 0.01$. *Language understanding*: Federated setup with $N = 20$, $q = 0.2$, Dirichlet $\beta = 0.8$. Clients run $K = 20$ local steps for $T = 100$ rounds ($\mathcal{B} = 16$). We use LoRA with $r = \alpha = 8$, freezing the classification head. AdamW optimizer with $\eta \in \{1e-4, 2e-4, 3e-4, 4e-4, 1e-3, 2e-3, 4e-3\}$. The privacy budget is also set to $\epsilon \in \{3, 2, 1\}$, with $\delta = 1e-5$ for SST-2 and QNLI, $\delta = 1e-6$ for QQP and MNLI, and smoothing $\sigma_s = 0.001$. We run 3 trials for vision tasks and 15 trials for language tasks, reporting the mean test accuracy across trials. Full details see Appendix B.2, C.2.

## 6.2 RESULTS

**Fine-tuning for image classification.** Table 3 and Figure 6 show that LA-LoRA consistently outperforms DP-LoRA, FFA-LoRA, and RoLoRA on CIFAR-100 and Tiny-ImageNet using both Swin-T and Swin-B, under privacy budgets $\epsilon \in \{3, 2, 1\}$. For Swin-T at $\epsilon = 3$, LA-LoRA achieves 60.07% (CIFAR-100) and 60.97% (Tiny-ImageNet), exceeding RoLoRA by **4.88%** and **10.10%**. Even at $\epsilon = 1$, LA-LoRA remains the top performer with 56.68% and 60.01% on the two datasets.

A similar trend holds for Swin-B. At $\epsilon = 1$, LA-LoRA reaches 74.56% (CIFAR-100) and 60.68% (Tiny-ImageNet), outperforming RoLoRA by **6.68%** and **16.83%**, respectively, further confirming its consistent superiority over baselines. Further experimental details are provided in Appendix B.3.

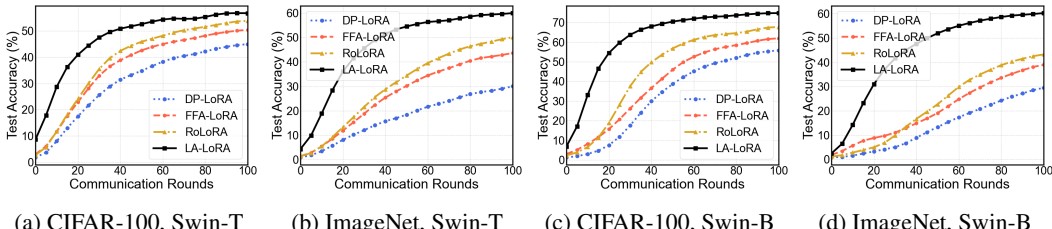

| (a) CIFAR-100, Swin-T | (b) ImageNet, Swin-T | (c) CIFAR-100, Swin-B | (d) ImageNet, Swin-B |

Figure 6: Test accuracy of Swin-T and Swin-B on CIFAR-100 and Tiny-ImageNet with $\epsilon = 1$.

Table 3: Test accuracy of Swin-T and Swin-B on CIFAR-100 and Tiny-ImageNet for different $\epsilon$.

| Privacy Budget | Method | Swin-T Model (%) | | Swin-B Model (%) | |
|---|---|---|---|---|---|
| | | CIFAR-100 | Tiny-ImageNet | CIFAR-100 | Tiny-ImageNet |
| $\epsilon = 3$ | DP-LoRA | $45.40_{\pm 0.40}$ | $32.27_{\pm 0.91}$ | $56.52_{\pm 0.51}$ | $30.64_{\pm 0.30}$ |
| | FFA-LoRA | $52.09_{\pm 0.36}$ | $44.62_{\pm 0.55}$ | $62.10_{\pm 0.39}$ | $39.84_{\pm 0.19}$ |
| | RoLoRA | $55.19_{\pm 0.42}$ | $50.87_{\pm 0.56}$ | $67.96_{\pm 0.64}$ | $44.18_{\pm 0.66}$ |
| | **LA-LoRA** | $\mathbf{60.07}_{\pm 0.41}$ | $\mathbf{60.97}_{\pm 0.44}$ | $\mathbf{75.29}_{\pm 0.35}$ | $\mathbf{61.97}_{\pm 0.56}$ |
| $\epsilon = 2$ | DP-LoRA | $44.82_{\pm 0.57}$ | $32.14_{\pm 1.39}$ | $56.31_{\pm 0.28}$ | $30.31_{\pm 0.55}$ |
| | FFA-LoRA | $52.05_{\pm 0.43}$ | $44.31_{\pm 0.44}$ | $62.02_{\pm 0.33}$ | $39.54_{\pm 0.49}$ |
| | RoLoRA | $55.02_{\pm 0.35}$ | $50.56_{\pm 0.57}$ | $67.93_{\pm 0.40}$ | $43.97_{\pm 0.38}$ |
| | **LA-LoRA** | $\mathbf{59.52}_{\pm 0.53}$ | $\mathbf{60.63}_{\pm 0.49}$ | $\mathbf{74.93}_{\pm 0.32}$ | $\mathbf{61.03}_{\pm 0.67}$ |
| $\epsilon = 1$ | DP-LoRA | $45.58_{\pm 0.47}$ | $31.00_{\pm 0.47}$ | $55.98_{\pm 0.56}$ | $30.20_{\pm 0.46}$ |
| | FFA-LoRA | $50.75_{\pm 0.54}$ | $44.38_{\pm 0.38}$ | $61.94_{\pm 0.37}$ | $39.33_{\pm 0.48}$ |
| | RoLoRA | $54.88_{\pm 0.66}$ | $50.78_{\pm 0.45}$ | $67.88_{\pm 0.32}$ | $43.85_{\pm 0.60}$ |
| | **LA-LoRA** | $\mathbf{56.68}_{\pm 0.60}$ | $\mathbf{60.01}_{\pm 0.51}$ | $\mathbf{74.56}_{\pm 0.52}$ | $\mathbf{60.68}_{\pm 0.55}$ |

**Fine-tuning for language understanding**. Table 4 summarizes the results, where LA-LoRA consistently outperforms all baselines across datasets and privacy levels. At $\epsilon = 3$, LA-LoRA achieves 93.12% on SST-2 and 89.83% on QNLI. At $\epsilon = 1$, it maintains 85.34% on QQP and 82.35% on MNLI, surpassing the best baseline RoLoRA in all cases. More results in Appendix C.3.

## 6.3 ABLATION STUDY

To better understand the individual contributions of the local alternating update strategy and the low-pass smoothing filter, we fixed $\epsilon = 3$ for ablation studies.

Table 4: Test accuracy (%) of RoBERTa-Base on SST-2, QNLI, QQP and MNLI under different $\epsilon$.

| Privacy Budget | Method | SST-2 | QNLI | QQP | MNLI |
|---|---|---|---|---|---|
| $\epsilon = 3$ | DP-LoRA | $92.36_{\pm 0.75}$ | $86.31_{\pm 0.32}$ | $84.56_{\pm 0.83}$ | $80.98_{\pm 0.44}$ |
| | FFA-LoRA | $92.32_{\pm 0.49}$ | $87.20_{\pm 0.37}$ | $85.12_{\pm 0.34}$ | $81.71_{\pm 0.69}$ |
| | RoLoRA | $92.70_{\pm 0.52}$ | $88.23_{\pm 0.49}$ | $85.35_{\pm 0.42}$ | $82.12_{\pm 0.44}$ |
| | **LA-LoRA** | $\mathbf{93.12}_{\pm 0.67}$ | $\mathbf{89.83}_{\pm 0.41}$ | $\mathbf{85.83}_{\pm 0.49}$ | $\mathbf{82.99}_{\pm 0.42}$ |
| $\epsilon = 2$ | DP-LoRA | $92.20_{\pm 0.64}$ | $86.03_{\pm 0.57}$ | $84.26_{\pm 0.57}$ | $80.62_{\pm 0.35}$ |
| | FFA-LoRA | $92.39_{\pm 0.51}$ | $87.30_{\pm 0.60}$ | $84.73_{\pm 0.47}$ | $81.94_{\pm 0.58}$ |
| | RoLoRA | $92.55_{\pm 0.40}$ | $87.08_{\pm 0.48}$ | $85.02_{\pm 0.53}$ | $82.01_{\pm 0.44}$ |
| | **LA-LoRA** | $\mathbf{93.00}_{\pm 0.70}$ | $\mathbf{89.18}_{\pm 0.51}$ | $\mathbf{85.64}_{\pm 0.58}$ | $\mathbf{82.87}_{\pm 0.52}$ |
| $\epsilon = 1$ | DP-LoRA | $90.71_{\pm 0.55}$ | $84.07_{\pm 0.55}$ | $83.48_{\pm 0.38}$ | $79.87_{\pm 0.79}$ |
| | FFA-LoRA | $91.06_{\pm 0.53}$ | $85.08_{\pm 0.53}$ | $84.30_{\pm 0.57}$ | $81.14_{\pm 0.59}$ |
| | RoLoRA | $92.32_{\pm 0.41}$ | $86.25_{\pm 0.46}$ | $84.49_{\pm 0.57}$ | $81.54_{\pm 0.50}$ |
| | **LA-LoRA** | $\mathbf{92.66}_{\pm 0.47}$ | $\mathbf{88.73}_{\pm 0.42}$ | $\mathbf{85.34}_{\pm 0.35}$ | $\mathbf{82.35}_{\pm 0.46}$ |

**Effect of local alternating updates.** We evaluate DP-LoRA with LA-LoRA(-filter), which uses alternating updates without the filter. Across both language and vision tasks, LA-LoRA(-filter) consistently improves performance over DP-LoRA. For example, in Table 5, accuracy on Tiny-ImageNet with Swin-B improves from 30.64% to 53.07%, indicating substantial gains in deep vision models. Similarly, in Table 5, accuracy on QNLI improves from 86.31% to 88.92%. These results demonstrate that alternating updates improve model utility under DP, achieving a better trade-off.

**Effect of low-pass smoothing filter.** We compare DP-LoRA vs. DP-LoRA(+filter) and LA-LoRA(-filter) vs. LA-LoRA. In both domains, the filter consistently delivers additional performance gains. For example, on Tiny-ImageNet (Swin-B, Table 5), DP-LoRA(+filter) increases the accuracy from 30.64% to 49.85% over DP-LoRA, while applying the filter to LA-LoRA(-filter) further improves the accuracy from 53.07% to 61.97%. The filter facilitates smoother updates that bias optimization toward flatter global solutions, improving both stability and generalization under DP.

Appendix presents supplementary experiments, including more results for Gaussian low-pass smoothing filter E, computational and memory cost B.4, and other ablation studies D.

Table 5: Impact of local alternating updates and low-pass smoothing filter on federated LoRA performance (%) across GLUE (RoBERTa-Base) and image classification (Swin-B) benchmarks.

| Method | GLUE tasks (RoBERTa-Base) | | | | Image Classification (Swin-B) | |
|---|---|---|---|---|---|---|
| | SST-2 | QNLI | QQP | MNLI | CIFAR-100 | Tiny-ImageNet |
| DP-LoRA | $92.36_{\pm 0.75}$ | $86.31_{\pm 0.32}$ | $84.56_{\pm 0.83}$ | $80.98_{\pm 0.44}$ | $56.52_{\pm 0.51}$ | $30.64_{\pm 0.30}$ |
| DP-LoRA(+filter) | $92.52_{\pm 0.55}$ | $87.06_{\pm 0.66}$ | $84.79_{\pm 0.42}$ | $81.43_{\pm 0.57}$ | $69.08_{\pm 0.52}$ | $49.85_{\pm 0.55}$ |
| LA-LoRA(-filter) | $92.74_{\pm 0.67}$ | $88.92_{\pm 0.50}$ | $84.98_{\pm 0.51}$ | $82.40_{\pm 0.53}$ | $70.38_{\pm 0.48}$ | $53.07_{\pm 0.60}$ |
| **LA-LoRA** | $\mathbf{93.12}_{\pm 0.67}$ | $\mathbf{89.83}_{\pm 0.41}$ | $\mathbf{85.83}_{\pm 0.49}$ | $\mathbf{82.99}_{\pm 0.42}$ | $\mathbf{75.29}_{\pm 0.35}$ | $\mathbf{61.97}_{\pm 0.56}$ |

# 7 DISCUSSION AND CONCLUSION

In this work, we propose LA-LoRA, a privacy-preserving framework for differentially private federated adaptation. By alternating local updates and applying a low-pass smoothing filter, LA-LoRA addresses three key challenges in DPFL: gradient coupling, noise amplification, and sharp aggregation bias. Experiments on vision and language tasks show consistent improvements in accuracy, stability, and privacy efficiency. Future work will extend LA-LoRA along three axes: (i) combining DP-LoRA adaptation with faster and more communication-efficient federated optimizers (Liu et al., 2024b; 2025b;c); (ii) improving robustness and generalization under stronger heterogeneity via averaging/flatness-aware principles (Liu et al., 2024a; 2025a;d); and (iii) scaling to larger foundation models, potentially leveraging preconditioned/second-order updates while mitigating preconditioner drift (Liu et al., 2026).

## ACKNOWLEDGMENTS

This work was supported in part by the National Natural Science Foundation of China (No. U24A20243, No. 62232013, No. 62302363), in part by the National Key Research and Development Program of China (No. 2024YFB3108700), and in part by the Program of China Scholarship Council (CSC).

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

## 8 ETHICS STATEMENT

This work adheres to the ICLR Code of Ethics. In this study, no human subjects or animal experimentation was involved. All datasets used, including **CIFAR-100, Tiny-ImageNet, SST-2, QNLI, QQP, MNLI**, were sourced in compliance with relevant usage guidelines, ensuring no violation of privacy. We have taken care to avoid any biases or discriminatory outcomes in our research process. No personally identifiable information was used, and no experiments were conducted that could raise privacy or security concerns. We are committed to maintaining transparency and integrity throughout the research process.

## 9 REPRODUCIBILITY STATEMENT

We have made every effort to ensure that the results presented in this paper are reproducible. All code and datasets have been made publicly available in an anonymous repository (`https://github.com/junkangLiu0/LA-LORA`) to facilitate replication and verification. The experimental setup, including training steps, model configurations, and hardware details, is described in detail in Section 6, Appendix B.2, and C.2.

We believe these measures will enable other researchers to reproduce our work and further advance the field.

APPENDIX

**LIST OF APPENDIX**

## A   MORE RELATED WORK

Recent advances in federated optimization and generalization provide complementary perspectives to our DP federated adaptation setting. On the optimization side, FedBCGD (Liu et al., 2024b) develops an accelerated block coordinate gradient descent framework for communication-efficient FL, while FedMuon (Liu et al., 2025b) further accelerates training via matrix orthogonalization. For large-model federated training, FedAdamW (Liu et al., 2025c) introduces an AdamW-style optimizer with improved communication efficiency and stability. Beyond optimization speed, generalization under strong heterogeneity has been studied by FedSWA (Liu et al., 2024a), which leverages stochastic weight averaging to enhance robustness. Meanwhile, the geometry of FL objectives has attracted growing attention: FedNSAM (Liu et al., 2025a) analyzes the consistency relationship between local and global flatness, and in the DPFL regime, DP-FedPGN (Liu et al., 2025d) explicitly penalizes gradient norms to encourage flatter minima. Finally, second-order and preconditioned methods in FL face the challenge of client-induced preconditioner drift; FedPAC (Liu et al., 2026) mitigates this drift to unlock the potential of second-order optimizers.

## B   ADDITIONAL DETAILS FOR IMAGE CLASSIFICATION

Section 6 outlines the main experimental configurations and results. For completeness, we summarize only the additional settings not previously described and further results.

### B.1   DATASETS AND MODELS

- CIFAR-100 (Krizhevsky, 2009) comprises 100 categories organized into 20 broader groups, containing 50,000 training images and 10,000 test images. Each image is a 32×32 pixel RGB color image. Every category includes 600 images, with 500 for training and 100 for testing. Tiny-ImageNet (Le & Yang, 2015) is a subset of the ImageNet dataset containing 200 object categories, totaling around 100,000 images. Each image is a 64×64 pixel RGB color image. Every category consists of 500 training images, 50 validation images, and 50 test images.

- Swin-T (Liu et al., 2021) is the smallest variant of the Swin Transformer, using a 4×4 patch embedding with a 96-dimensional embedding size. It has stage depths of [2, 2, 6, 2], about 28M parameters. Swin-B (Liu et al., 2021) is a larger variant with a 4×4 patch embedding and a 128-dimensional embedding size. It has stage depths of [2, 2, 18, 2], about 88M parameters.

### B.2   EXPERIMENTAL SETUP

In image classification fine-tuning tasks, we apply LoRA with rank $r = \alpha = 16$, keeping the classification head trainable so that it can adapt to the target dataset's label space and feature distribution, and thus remain compatible with the LoRA-updated representations during local training.

We fix $\delta = 1e - 5$ and use an RDP accountant. For each target privacy budget $\epsilon \in \{3, 2, 1\}$, we grid-search the Rényi order $\lambda$ under the subsampled Gaussian mechanism with per-sample $\ell_2$ clipping, and choose the noise scale $\sigma$ that satisfies the $(\epsilon, \delta)$ constraint after converting from RDP. The resulting $\sigma$ are:

- **CIFAR-100**: $\sigma \in \{0.195, \ 0.29, \ 0.56\}$,

- **Tiny-ImageNet**: $\sigma \in \{0.098, \ 0.146, \ 0.283\}$.

These correspond to privacy budgets of $\epsilon \in \{3, \ 2, \ 1\}$ respectively. Gradient clipping is performed on a per-layer basis using a *median clipping* strategy, where each layer's clipping threshold $C$ is set to the median of its gradient norm distribution, enabling balanced sensitivity control and stable training under differential privacy constraints.

### B.3 ADDITIONAL EXPERIMENTAL RESULTS

Table 3 in Section 6 presents the final performance comparison between our LA-LoRA and three SOTA baselines across different $\epsilon$ values. In this section, we present the complete convergence curves corresponding to these experiments. Figure 7 presents the convergence curves for the Swin-T and Swin-B models under $\epsilon=2$. In both cases, LA-LoRA consistently outperforms the three SOTA baselines. The improvement is particularly pronounced on Tiny-ImageNet with Swin-B, where LA-LoRA surpasses RoLoRA's $43.97\%$ by $17.06\%$. Figure 8 illustrates the convergence of test accuracy for CIFAR-100 and Tiny-ImageNet using Swin-T and Swin-B under $\epsilon = 3$. LA-LoRA consistently achieves higher accuracy and faster convergence than the three baselines.

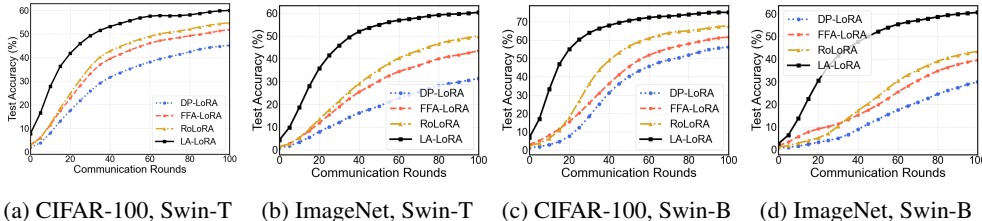

   (a) CIFAR-100, Swin-T    (b) ImageNet, Swin-T    (c) CIFAR-100, Swin-B    (d) ImageNet, Swin-B

Figure 7: Test accuracy of Swin-T and Swin-B on CIFAR-100 and Tiny-ImageNet with $\epsilon = 2$.

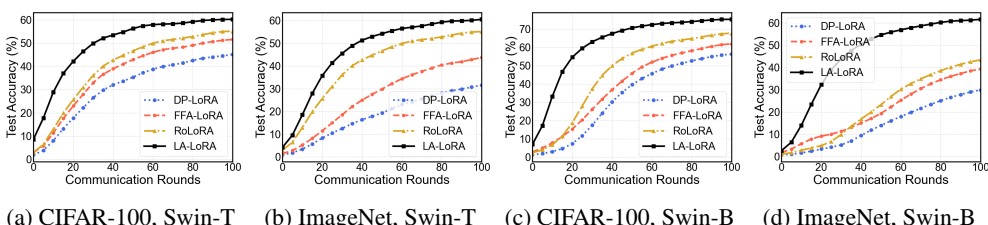

   (a) CIFAR-100, Swin-T    (b) ImageNet, Swin-T    (c) CIFAR-100, Swin-B    (d) ImageNet, Swin-B

Figure 8: Test accuracy of Swin-T and Swin-B on CIFAR-100 and Tiny-ImageNet with $\epsilon = 3$.

### B.4 COMPUTATIONAL AND MEMORY OVERHEAD

Table 6 reports the per-round computation cost, memory cost and test accuracy of Swin-B at $\epsilon = 1$. Compared with standard DP-LoRA, LA-LoRA reduces the per-round time from $30.35$ s to $17.44$ s on CIFAR-100 and from $28.02$ s to $17.23$ s on Tiny-ImageNet, and halves the memory cost from $3524$ MB to $1762$ MB. LA-LoRA updates only one LoRA factor at each local step, reducing the peak memory required for per-sample gradient computation.

Among LoRA-based DP baselines, LA-LoRA has a per-round time that is comparable to the fastest method: it differs by at most about 1 s, while consistently achieving the best accuracy. Overall, LA-LoRA offers substantial accuracy gains with negligible additional computational overhead compared to other low-rank DP methods.

Table 6: Per-round computation cost, memory cost and performance comparison of Swin-B at $\epsilon = 1$.

| Method | Time Cost (s) | | Memory Cost (MB) | | Test Accuracy (%) | |
|---|---|---|---|---|---|---|
| | CIFAR-100 | Tiny-ImageNet | CIFAR-100 | Tiny-ImageNet | CIFAR-100 | Tiny-ImageNet |
| DP-LoRA | 30.35 | 28.02 | 3524 | 3524 | 55.98 | 30.20 |
| DP-LoRA(+filter) | 30.72 | 28.51 | 3524 | 3524 | 67.95 | 48.09 |
| FFA-LoRA | 17.85 | 16.54 | 1762 | 1762 | 61.94 | 39.33 |
| RoLoRA | 16.64 | 16.32 | 1762 | 1762 | 67.88 | 43.85 |
| LA-LoRA(-filter) | 17.30 | 17.16 | 1762 | 1762 | 69.87 | 52.72 |
| **LA-LoRA** | 17.44 | 17.23 | 1762 | 1762 | 74.56 | 60.68 |

Table 7: Impact of rank of Swin-B on CIFAR-100 and Tiny-ImageNet at $\epsilon = 1$, averaged over 5 runs.

| Rank | Method | CIFAR-100 | Tiny-ImageNet |
|---|---|---|---|
| $r = 8$ | DP-LoRA | $54.30_{\pm 0.52}$ | $29.57_{\pm 0.53}$ |
| | FFA-LoRA | $59.78_{\pm 0.44}$ | $36.58_{\pm 0.35}$ |
| | RoLoRA | $65.74_{\pm 0.61}$ | $40.89_{\pm 0.56}$ |
| | **LA-LoRA** | $\mathbf{72.88}_{\pm 0.57}$ | $\mathbf{59.03}_{\pm 0.45}$ |
| $r = 16$ | DP-LoRA | $55.98_{\pm 0.56}$ | $30.20_{\pm 0.46}$ |
| | FFA-LoRA | $61.94_{\pm 0.37}$ | $39.33_{\pm 0.48}$ |
| | RoLoRA | $67.88_{\pm 0.32}$ | $43.85_{\pm 0.60}$ |
| | **LA-LoRA** | $\mathbf{74.56}_{\pm 0.52}$ | $\mathbf{60.68}_{\pm 0.55}$ |
| $r = 32$ | DP-LoRA | $56.33_{\pm 0.28}$ | $31.57_{\pm 0.55}$ |
| | FFA-LoRA | $62.64_{\pm 0.33}$ | $44.45_{\pm 0.37}$ |
| | RoLoRA | $67.97_{\pm 0.40}$ | $44.36_{\pm 0.39}$ |
| | **LA-LoRA** | $\mathbf{75.34}_{\pm 0.51}$ | $\mathbf{62.99}_{\pm 0.42}$ |

## B.5 IMPACT OF DIFFERENT RANKS ON PERFORMANCE

Table 7 summarizes the test accuracy of four methods on the Swin-B model for CIFAR-100 and Tiny-ImageNet under a privacy budget of $\epsilon{=}1$ with varying rank values ($r{=}8, 16, 32$). LA-LoRA consistently achieves the highest accuracy in all configurations, attaining $75.34\%$ on CIFAR-100 and $62.99\%$ on Tiny-ImageNet when $r{=}32$. In contrast, DP-LoRA exhibits the lowest accuracy and the greatest sensitivity to rank variation, indicating that its performance is more constrained by representational capacity under strict privacy constraints. FFA-LoRA and RoLoRA demonstrate intermediate performance, benefiting steadily from increased rank. Furthermore, Tiny-ImageNet is complex, resulting in slower convergence within the limited number of communication rounds. Consequently, its absolute accuracy is substantially lower than that of CIFAR-100, while the relative ranking of the methods remains unchanged.

These observations indicate that increasing the rank can enhance representational capacity and mitigate the adverse effects of differential privacy noise, with LA-LoRA exploiting this advantage most effectively. However, a larger rank also introduces potential drawbacks, including increased communication cost, heavier local computation, and a higher risk of overfitting under limited communication rounds. Balancing these trade-offs, we adopt $r = 16$ as the baseline setting in our experiments to achieve a favorable compromise between performance and efficiency.

## B.6 CENTRALIZED EXPERIMENTS

To verify how the challenges discussed in Section 3 manifest in the centralized setting, we conduct centralized DP experiments using the same model architecture, optimizer, clipping norm, and noise multiplier as in our federated setup.

As summarized in Table 8, LA-LoRA(-filter) achieves higher test accuracy and higher "Grad. Cos. (late)" than DP-LoRA in the centralized setting. This confirms that the DP–LoRA issues we identify are not specific to federated training. Moreover, the gain of LA-LoRA(-filter) over DP-LoRA increases from $1.18\%$ in centralized DP training to $11.22\%$ in federated DP training, and the gradient cosine gap widens from $0.032$ to $0.108$, indicating that federated optimization exacerbates the gradient coupling inherent in centralized settings.

## B.7 NON-PRIVATE FEDERATED EXPERIMENTS

To assess whether the observed gradient behavior persists in the absence of DP, we run federated experiments *without* DP. Table 9 shows that, even without gradient clipping and $\sigma = 0$, LA-LoRA(-filter) achieves higher test accuracy and higher "Grad. Cos. (late)" than Fed-LoRA, indicating that gradient alignment is beneficial even in the non-private setting.

Table 8: Centralized and federated DP training for Swin-T on CIFAR-100 with $\epsilon = 3$. "Grad. Cos. (late)" denotes the average cosine similarity between $\nabla_A \mathcal{L}$ and $\nabla_B \mathcal{L}$ over the last 10% of training steps. Federated DP-LoRA has lower test accuracy and gradient cosine than centralized DP-LoRA, while LA-LoRA(-filter) improves both settings.

| Setting | Method | Test Acc. (%) | $\Delta$Acc | Grad. Cos. (late) | $\Delta$Cos |
|---|---|---|---|---|---|
| Centralized | DP-LoRA | $76.11_{\pm 0.38}$ | - | 0.681 | - |
| | LA-LoRA(-filter) | $77.29_{\pm 0.43}$ | $\uparrow 1.18$ | 0.713 | $\uparrow 0.032$ |
| Federated | DP-LoRA | $45.40_{\pm 0.40}$ | - | 0.337 | - |
| | LA-LoRA(-filter) | $56.62_{\pm 0.54}$ | $\uparrow 11.22$ | 0.445 | $\uparrow 0.108$ |

Table 9: Non-private federated training for Swin-T on CIFAR-100. "Grad. Cos. (late)" denotes the average cosine similarity between $\nabla_A \mathcal{L}$ and $\nabla_B \mathcal{L}$ over the last 10% of training steps.

| | Method | Test Acc. (%) | $\Delta$Acc | Grad. Cos. (late) | $\Delta$Cos |
|---|---|---|---|---|---|
| Non-private | Fed-LoRA | $90.56_{\pm 0.22}$ | - | 0.694 | - |
| | LA-LoRA(-filter) | $91.25_{\pm 0.15}$ | $\uparrow 0.69$ | 0.783 | $\uparrow 0.089$ |

Table 10 shows the performance comparison on Swin-B under a non-private federated architecture. DP-LoRA is the same as Fed-LoRA. On the CIFAR-100, LA-LoRA achieves a test accuracy of 84.21%, about 1% higher than DP-LoRA (83.21%) and RoLoRA (83.25%).

Table 10: Non-private federated training for Swin-B on CIFAR-100 and Tiny-ImageNet.

| | Method | CIFAR-100 | Tiny-ImageNet |
|---|---|---|---|
| Non-private | DP-LoRA | 83.21 | 81.37 |
| | FFA-LoRA | 81.65 | 80.76 |
| | RoLoRA | 83.25 | 81.03 |
| | **LA-LoRA** | **84.21** | **82.58** |

## C  ADDITIONAL DETAILS FOR LANGUAGE UNDERSTANDING

### C.1  DATASETS AND MODELS

- We evaluate our approach on four representative tasks from the GLUE benchmark (Wang et al., 2018): SST-2, QNLI, QQP, and MNLI. SST-2 is a binary sentiment classification dataset derived from the Stanford Sentiment Treebank, containing about 67K training sentences. QNLI is a binary natural language inference dataset converted from the Stanford Question Answering Dataset (SQuAD), with approximately 105K training sentence–question pairs. QQP is a binary paraphrase identification dataset from Quora, comprising around 364K training question pairs. MNLI is a three-way natural language inference dataset with multi-genre coverage, containing roughly 393K training sentence pairs.

- RoBERTa-Base (Liu et al., 2019) is a transformer-based language model optimized for robust pretraining. It adopts the BERT architecture with 12 transformer encoder layers, 12 self-attention heads per layer, and a hidden size of 768, totaling approximately 125M parameters. Compared to the original BERT, RoBERTa removes the next-sentence prediction objective, uses larger batch sizes, trains on more data, and applies dynamic masking, resulting in improved performance across a variety of NLP benchmarks.

### C.2  EXPERIMENTAL SETUP

For language understanding tasks, we freeze the classification head to preserve the well-trained label mapping of the language models, reduce the susceptibility of its relatively small parameter space to DP noise, and ensure stable and efficient adaptation via LoRA updates. We apply LoRA with $r = \alpha = 8$ to match the language models, using a maximum sequence length of $l_{\text{seq}} = 128$.

For privacy parameters, we set $\delta = 1e-5$ for SST-2 and QNLI, and $\delta = 1e-6$ for QQP and MNLI to account for their larger dataset sizes. The noise multipliers corresponding to privacy budgets $\epsilon \in \{3, 2, 1\}$ are:

- **SST-2**: $\sigma \in \{0.36, 0.53, 1.0\}$,
- **QNLI**: $\sigma \in \{0.23, 0.34, 0.67\}$,
- **QQP**: $\sigma \in \{0.073, 0.11, 0.21\}$,
- **MNLI**: $\sigma \in \{0.067, 0.10, 0.195\}$.

Gradient clipping follows the same strategy as in the image classification setup.

## C.3 ADDITIONAL EXPERIMENTAL RESULTS

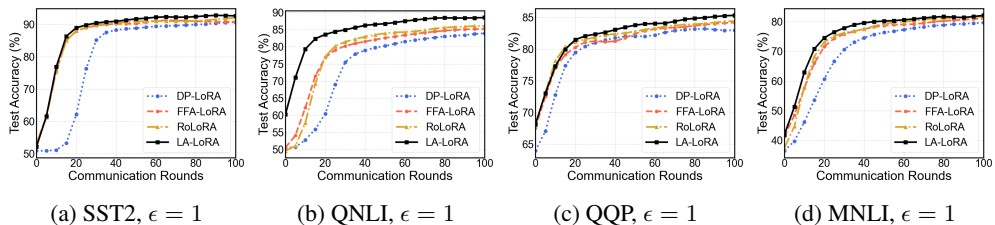

(a) SST2, $\epsilon = 1$      (b) QNLI, $\epsilon = 1$      (c) QQP, $\epsilon = 1$      (d) MNLI, $\epsilon = 1$

Figure 9: Test accuracy of RoBERTa-Base on SST-2, QNLI, QQP, and MNLI with $\epsilon = 1$.

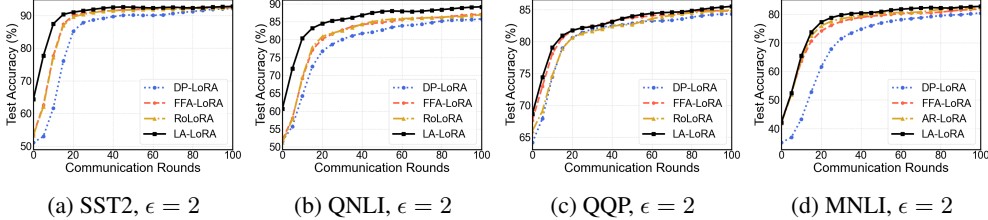

(a) SST2, $\epsilon = 2$      (b) QNLI, $\epsilon = 2$      (c) QQP, $\epsilon = 2$      (d) MNLI, $\epsilon = 2$

Figure 10: Test accuracy of RoBERTa-Base on SST-2, QNLI, QQP, and MNLI with $\epsilon = 2$.

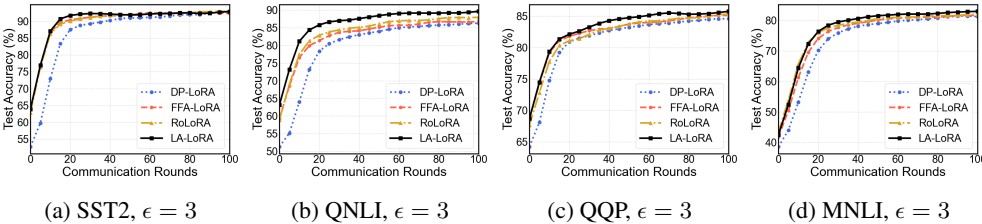

(a) SST2, $\epsilon = 3$      (b) QNLI, $\epsilon = 3$      (c) QQP, $\epsilon = 3$      (d) MNLI, $\epsilon = 3$

Figure 11: Test accuracy of RoBERTa-Base on SST-2, QNLI, QQP, and MNLI with $\epsilon = 3$.

Figure 9, Figure 10, and Figure 11 present the convergence curves of LA-LoRA and three SOTA baselines (DP-LoRA, FFA-LoRA, RoLoRA) on SST-2, QNLI, QQP, and MNLI using RoBERTa-Base under different privacy budgets ($\epsilon \in \{1, 2, 3\}$). Table 4 summarizes the corresponding final test accuracies. Across all settings, LA-LoRA consistently achieves the highest accuracy while maintaining fast convergence.

When $\epsilon = 1$ (Figure 9), where the privacy is the strongest, the performance gap between LA-LoRA and the baselines is the most pronounced. For instance, on QNLI, LA-LoRA reaches $88.73\%$ compared with $86.25\%$ for RoLoRA and $85.08\%$ for FFA-LoRA, showing improvements of $2.48\%$ and $3.65\%$, respectively. Similar gains are observed for QQP ($0.85\%$ over RoLoRA, $1.04\%$ over FFA-LoRA) and MNLI ($0.81\%$ over RoLoRA, $1.21\%$ over FFA-LoRA).

At $\epsilon = 2$ (Figure 10), the accuracy gap narrows, but LA-LoRA still leads in all datasets. For example, on QNLI, LA-LoRA obtains $89.18\%$, exceeding RoLoRA ($87.08\%$) and FFA-LoRA ($87.30\%$) by roughly $2.10\%$ and $1.88\%$.

At $\epsilon = 3$ (Figure 11), where privacy constraints are weakest, all methods perform more closely, yet LA-LoRA maintains the highest accuracy across all datasets.

### C.4 ADDITIONAL EXPERIMENTS ON LLAMA-2-7B

We evaluate our method on the Llama-2-7B model on a subset of the GLUE benchmark (MNLI, QNLI, QQP, SST-2). We follow exactly the same fine-tuning hyperparameters and training pipeline as in our main experiments, changing only the underlying backbone model.

Table 11 shows that, compared to baseline DP-LoRA, FFA-LoRA, RoLoRA, and LA-LoRA yield consistent gains in the four GLUE tasks. Among them, LA-LoRA achieves the best average performance, improving the DP-LoRA baseline by $1.31\%$. Moreover, LA-LoRA further outperforms the strongest baseline variant, RoLoRA, by $0.7\%$. The improvements are noticeable on QQP and MNLI, indicating that our approach may also transfer to a larger, more recent LLM backbone.

Table 11: Test accuracy of Llama-2-7B on SST-2, QNLI, QQP and MNLI with $\epsilon = 1$.

| Model | Method | SST-2 | QNLI | QQP | MNLI | Avg |
|---|---|---|---|---|---|---|
| | DP-LoRA | 91.56 | 88.22 | 85.56 | 86.86 | 88.05 |
| | FFA-LoRA | 92.53 | 89.23 | 85.56 | 86.98 | 88.58 |
| Llama-2-7B | RoLoRA | 92.12 | 89.34 | 85.98 | 87.21 | 88.66 |
| | LA-LoRA | 93.36 | 89.78 | 86.75 | 87.56 | 89.36 |

### C.5 LANGUAGE MODEL RESULTS UNDER DATA HETEROGENEITY $\beta = 0.3$

We report additional results of language tasks under Dirichlet $\beta = 0.3$. Table 12 summarizes the performance of all baselines and LA-LoRA variants on GLUE tasks.

Table 12: Test accuracy(%) of RoBERTa-Base on SST-2, QNLI, QQP and MNLI, Dirichlet $\beta = 0.3$.

| Privacy | Method | SST-2 | QNLI | QQP | MNLI | Avg. |
|---|---|---|---|---|---|---|
| | DP-LoRA | 92.07 | 86.25 | 84.02 | 81.22 | 85.89 |
| | FFA-LoRA | 92.56 | 87.53 | 85.36 | 81.54 | 86.75 |
| Non-private | RoLoRA | 93.65 | 88.65 | 85.52 | 82.22 | 87.51 |
| | **LA-LoRA** | **93.94** | **89.96** | **86.41** | **83.32** | **88.41** |
| | DP-LoRA | 90.13 | 83.79 | 83.28 | 79.80 | 84.25 |
| | FFA-LoRA | 90.62 | 84.63 | 84.10 | 80.98 | 85.08 |
| $\epsilon = 1$ | RoLoRA | 91.74 | 85.86 | 84.21 | 81.35 | 85.78 |
| | **LA-LoRA** | **92.11** | **87.13** | **85.04** | **82.27** | **86.64** |

When $\epsilon = 1$, LA-LoRA achieves the best average score on GLUE, improving over DP-LoRA by $2.39\%$ and over the best alternating baseline RoLoRA by $0.86\%$. In the non-private setting, LA-LoRA improves average accuracy by $2.52\%$ over DP-LoRA and by $0.90\%$ over the best alternating baseline, showing that LA-LoRA is beneficial even without DP noise.

## D ADDITIONAL ABLATION STUDIES

### D.1 ABLATION RESULTS FOR SWIN-T ON CIFAR-100 AND TINY-IMAGENET

Table 13 reports the ablation results for Swin-T on CIFAR-100 and Tiny-ImageNet, examining the individual contributions of the local alternating update strategy and the low-pass smoothing filter. Consistent with the findings in the main text for Swin-B (Table 5), applying local alternating updates (DP-LoRA $\rightarrow$ LA-LoRA(-filter)) substantially improves accuracy over the baseline DP-LoRA, with

gains of 11.22% on CIFAR-100 and 19.91% on Tiny-ImageNet. Furthermore, incorporating the low-pass smoothing filter (LA-LoRA(-filter) → LA-LoRA) delivers additional performance boosts, reaching 60.07% and 60.97% on CIFAR-100 and Tiny-ImageNet, respectively. These results confirm that both components contribute positively and complementarily to performance, with the combination yielding the highest accuracy across datasets.

Table 13: Effect of local alternating updates and low-pass smoothing filter for Swin-T, $\epsilon = 3$.

| Method | CIFAR-100 | Tiny-ImageNet |
|---|---|---|
| DP-LoRA | $45.40_{\pm 0.40}$ | $32.27_{\pm 0.91}$ |
| DP-LoRA(+filter) | $55.75_{\pm 0.63}$ | $50.69_{\pm 0.67}$ |
| LA-LoRA(-filter) | $56.62_{\pm 0.54}$ | $52.18_{\pm 0.37}$ |
| **LA-LoRA** | $\mathbf{60.07}_{\pm 0.41}$ | $\mathbf{60.97}_{\pm 0.44}$ |

## D.2 ABLATION ON SMOOTHING STRATEGIES

Recent work has proposed Doppler (Zhang et al., 2024b) as a generic low-pass filtering module for differentially private optimizers, post-processing privatized gradients in standard (non-federated) DP training to improve their signal-to-noise ratio.

To place our simple Gaussian filter in context, we instantiate Doppler in our DPFL setting with LoRA and treat it as a baseline. Specifically, for both DP-LoRA and LA-LoRA we compare three variants under exactly the same DPFL hyperparameters: no smoothing, Doppler, and our Gaussian filter, all applied on top of the same privatized local LoRA updates. All other optimizer, model, and privacy parameters are kept fixed. For Doppler, we evaluate every configuration reported in Table 2 of Zhang et al. (2024b) and find $b_\tau = \{1, 1\}/11$ and $a_\tau = -9/11$ to perform best in our setting.

Table 15 reports the results. On GLUE with RoBERTa-Base, both low-pass filters bring small but consistent gains over DP-LoRA: LA-LoRA improves QQP and MNLI from 84.56%/80.98% to 85.83%/82.99% (+1.27% / +2.01%), slightly outperforming Doppler 85.45%/82.13%. On the more challenging vision benchmarks, the effect is substantially larger. For Swin-B, our Gaussian low-pass filter raises DP-LoRA accuracy from 56.52% to 69.08% on CIFAR-100 (+12.56%) and from 30.64% to 49.85% on Tiny-ImageNet (+19.21%), whereas Doppler reaches 66.12% (+9.60%) and 48.53% (+17.89%), respectively. Combining the filter with local alternating updates, LA-LoRA further improves CIFAR-100 and Tiny-ImageNet to 75.29% and 61.97%, outperforming the corresponding LA-LoRA(+Doppler) variant by 2.41% and 4.61%.

Table 14: Impact of different low-pass filters on federated LoRA performance (%) across GLUE and image classification benchmarks, $\epsilon = 3$.

| Method | GLUE tasks (RoBERTa-Base) | | Image Classification (Swin-B) | |
|---|---|---|---|---|
| | QQP | MNLI | CIFAR-100 | Tiny-ImageNet |
| DP-LoRA | $84.56_{\pm 0.83}$ | $80.98_{\pm 0.44}$ | $56.52_{\pm 0.51}$ | $30.64_{\pm 0.30}$ |
| DP-LoRA(+filter) | $84.79_{\pm 0.42}$ | $81.43_{\pm 0.57}$ | $69.08_{\pm 0.52}$ | $49.85_{\pm 0.55}$ |
| DP-LoRA(+Doppler) | $84.82_{\pm 0.49}$ | $81.04_{\pm 0.57}$ | $66.12_{\pm 0.71}$ | $48.53_{\pm 0.57}$ |
| LA-LoRA(-filter) | $84.98_{\pm 0.51}$ | $82.40_{\pm 0.53}$ | $70.38_{\pm 0.48}$ | $53.07_{\pm 0.60}$ |
| **LA-LoRA** | $\mathbf{85.83}_{\pm 0.49}$ | $\mathbf{82.99}_{\pm 0.42}$ | $\mathbf{75.29}_{\pm 0.35}$ | $\mathbf{61.97}_{\pm 0.56}$ |
| LA-LoRA(+Doppler) | $85.45_{\pm 0.50}$ | $82.13_{\pm 0.58}$ | $72.88_{\pm 0.56}$ | $57.36_{\pm 0.66}$ |

Overall, our ablations indicate that both Doppler and our Gaussian filter improve DPFL with LoRA by smoothing privatized updates, but in different ways. Doppler uses a recursive filter that depends on past outputs and tuned coefficients, whereas our Gaussian filter is a short window weighted average with fixed weights. In our setting, the privatized low rank updates exhibit substantial high frequency noise across local steps and clients. This simple Gaussian kernel helps suppress these high frequency fluctuations and, on our benchmarks, yields larger gains than Doppler, especially on the more challenging vision tasks.

## D.3 ABLATION ON THE THREE CHALLENGES

In this subsection, we empirically decompose how the two components of LA-LoRA (*locally alternating updates* and the *low-pass filter*) relate to the three challenges identified in Sec. 3: gradient coupling, noise amplification, and loss sharpness. We report the Maximum Hessian eigenvalue $\lambda_{\max}(H)$ restricted to LoRA parameters.

Table 15: Ablation of LA-LoRA components and the three challenges (noise amplification, gradient coupling, and loss sharpness) on GLUE ($\beta = 0.3$) and image classification ($\beta = 0.1$), $\epsilon = 1$. We report test accuracy (%) and the maximum Hessian eigenvalue $\lambda_{\max}(H)$ on LoRA parameters.

| Method | GLUE tasks (RoBERTa-Base) | | Image Classification (Swin-B) | |
|---|---|---|---|---|
| | QQP | $\lambda_{\max}(H)$ | CIFAR-100 | $\lambda_{\max}(H)$ |
| DP-LoRA | 84.02 | 43.74 | 55.98 | 101.62 |
| DP-LoRA(+filter) | 85.63 | $41.36_{(\downarrow 2.38)}$ | 67.95 | $80.33_{(\downarrow 21.29)}$ |
| LA-LoRA(-filter) | 85.95 | $40.82_{(\downarrow 2.92)}$ | 69.87 | $64.77_{(\downarrow 36.85)}$ |
| **LA-LoRA** | **86.41** | $40.22_{(\downarrow 3.52)}$ | **74.56** | $55.76_{(\downarrow 45.86)}$ |

Comparing **DP-LoRA** with **DP-LoRA(+filter)** isolates the effect of the low-pass filter. The accuracy gains (55.98% to 67.95% on CIFAR-100) and the drop in ($\lambda_{\max}(H)$ from 101.62 to 80.33) show that suppressing noise amplification already improves utility and smooths the loss landscape.

Comparing **DP-LoRA** with **LA-LoRA(-filter)** instead isolates the effect of locally alternating $B$ and $A$ at the same noise level. The larger improvement in both accuracy (55.98% to 69.87%, $\lambda_{\max}(H)$ from 101.62 to 64.77), indicating that mitigating gradient coupling is particularly important for deep vision backbones and contributes strongly to reducing loss sharpness.

LA-LoRA combines both components. Relative to all ablated variants, it achieves the highest accuracy on both QQP and CIFAR-100 and the smallest Maximum Hessian eigenvalue (e.g., $\lambda_{\max}(H)$ from 101.62 to 55.76), showing that jointly addressing noise amplification and gradient coupling drives the model towards significantly flatter minima.

## D.4 EFFECT OF LOCAL STEPS K

We study how the local step number $K$ affects LA-LoRA under a fixed clipping norm, noise multiplier, and number of communication rounds. In this setting, increasing $K$ makes each client perform more noisy local updates, so by standard DP composition the overall privacy loss $\epsilon$ grows with $K$. As shown in Table 16, larger $K$ generally leads to higher test accuracy on both CIFAR-100 and Tiny-ImageNet. At the same time, larger $K$ also incurs higher client computation and a larger privacy budget. To balance model quality, privacy, and training cost, we therefore set $K = 20$ as the default choice in all main experiments.

Table 16: LA-LoRA test accuracy (%) with different local steps $K$ (Swin-B).

| $K$ | $\sigma$ | 10 | 20 | 30 | 50 |
|---|---|---|---|---|---|
| CIFAR-100(%) | $\sigma = 0.56$ | 72.13 | 74.56 | 75.21 | 75.26 |
| Tiny-ImageNet(%) | $\sigma = 0.283$ | 58.54 | 60.68 | 61.24 | 61.32 |

## D.5 EFFECT OF DIFFERENT LOCAL ALTERNATING UPDATE STRATEGIES

We vary the local alternating strategies between the two LoRA factors while keeping the total local steps fixed. Table 17 shows nearly identical results across strategies: the best-worst gap is always below 0.7% (typically within 0.3%). Default *1-step B / 1-step A* strategy attains the best accuracy in most cases. These results indicate that LA-LoRA is insensitive to the precise local alternating strategy.

Table 17: Effect of different local alternating strategies on CIFAR-100 and Tiny-ImageNet (Swin-B) under $\epsilon = 3$ and different Dirichlet distributions. Strategy "$k$-step $B$ / $k$-step $A$" denotes performing $k$ local gradient steps on $B$ followed by $k$ steps on $A$ in each local round.

| Strategy | Dirichlet $\beta = 0.6$ | | Dirichlet $\beta = 0.1$ | |
|---|---|---|---|---|
| | CIFAR-100 | Tiny-ImageNet | CIFAR-100 | Tiny-ImageNet |
| 1-step B / 1-step A (default) | $89.62_{\pm 0.37}$ | $80.77_{\pm 0.50}$ | $75.29_{\pm 0.35}$ | $61.97_{\pm 0.56}$ |
| 2-step B / 2-step A | $89.29_{\pm 0.47}$ | $80.58_{\pm 0.39}$ | $75.15_{\pm 0.38}$ | $61.36_{\pm 0.42}$ |
| 5-step B / 5-step A | $89.31_{\pm 0.42}$ | $80.72_{\pm 0.40}$ | $75.47_{\pm 0.46}$ | $61.78_{\pm 0.51}$ |
| 5-step A / 5-step B | $89.45_{\pm 0.26}$ | $80.70_{\pm 0.42}$ | $75.34_{\pm 0.44}$ | $61.79_{\pm 0.58}$ |

# E    MORE RESULTS FOR GAUSSIAN LOW-PASS SMOOTHING FILTER

## E.1    SMOOTHING PARAMETER $\sigma_s$

Table 18 presents the performance of federated LoRA under different smoothing parameters $\sigma_s$ on GLUE (RoBERTa-Base) language understanding tasks and Swin-B image classification benchmarks ($\epsilon = 1$). For language tasks, relatively small values of $\sigma_s$ (e.g., $\sigma_s$=0.001) yield consistently strong and stable performance across datasets, suggesting that mild smoothing effectively suppresses local update noise while preserving task-relevant information. In contrast, for image classification tasks, moderately larger values (e.g., $\sigma_s$=0.01) tend to produce more robust results, indicating that stronger smoothing can better mitigate the impact of data heterogeneity and noisy updates in vision settings.

Table 18: Impact of the smoothing parameter $\sigma_s$ on federated LoRA performance (%) across GLUE (RoBERTa-Base) language tasks and image classification (Swin-B) benchmarks ($\epsilon = 1$).

| Smoothing parameter$\sigma_s$ | GLUE tasks (RoBERTa-Base) | | | | Image Classification (Swin-B) | |
|---|---|---|---|---|---|---|
| | SST-2 | QNLI | QQP | MNLI | CIFAR-100 | Tiny-ImageNet |
| $\sigma_s = 0.000$ | $92.51_{\pm 0.45}$ | $87.95_{\pm 0.56}$ | $84.72_{\pm 0.49}$ | $81.83_{\pm 0.55}$ | $70.34_{\pm 0.32}$ | $54.05_{\pm 0.52}$ |
| $\sigma_s = 0.001$ | $\mathbf{92.66}_{\pm 0.47}$ | $88.73_{\pm 0.42}$ | $\mathbf{85.34}_{\pm 0.35}$ | $82.35_{\pm 0.46}$ | $71.80_{\pm 0.26}$ | $56.14_{\pm 0.45}$ |
| $\sigma_s = 0.005$ | $92.60_{\pm 0.44}$ | $\mathbf{88.75}_{\pm 0.53}$ | $85.26_{\pm 0.42}$ | $\mathbf{82.42}_{\pm 0.55}$ | $72.67_{\pm 0.49}$ | $58.75_{\pm 0.38}$ |
| $\sigma_s = 0.010$ | $92.62_{\pm 0.47}$ | $87.97_{\pm 0.42}$ | $85.03_{\pm 0.48}$ | $82.03_{\pm 0.32}$ | $\mathbf{74.56}_{\pm 0.52}$ | $60.68_{\pm 0.55}$ |
| $\sigma_s = 0.050$ | $92.50_{\pm 0.49}$ | $88.02_{\pm 0.54}$ | $84.96_{\pm 0.36}$ | $82.03_{\pm 0.57}$ | $74.33_{\pm 0.40}$ | $\mathbf{61.27}_{\pm 0.56}$ |

The results further indicate that the optimal range of $\sigma_s$ is task-dependent. Values between 0.001 and 0.005 are generally favorable for language tasks, whereas values between 0.01 and 0.05 are more suitable for vision tasks, offering a better balance between stability and accuracy in each domain.

## E.2    EFFECT OF DIFFERENT KERNEL WIDTHS

As described in Section 4.2, LA-LoRA employs a simple 1D low-pass filter to smooth the LoRA gradients before aggregation. In the main experiments, we use a 5-tap binomial Gaussian kernel as the default choice. To assess the robustness of LA-LoRA, we conduct a sensitivity study on different kernel widths. We consider three 1D binomial Gaussian kernels with different sizes:

$$G_s^{(3)} = \tfrac{1}{4}[1, 2, 1], \quad G_s^{(5)} = \tfrac{1}{16}[1, 4, 6, 4, 1], \quad G_s^{(7)} = \tfrac{1}{64}[1, 6, 15, 20, 15, 6, 1].$$

These correspond to 3-, 5-, and 7-tap binomial filters, respectively, obtained from the binomial coefficients of $(1+1)^2$, $(1+1)^4$, and $(1+1)^6$ and normalized to sum to 1. Intuitively, larger kernels apply stronger smoothing, whereas smaller kernels apply milder smoothing.

Table 19 summarizes the results for Swin-B on CIFAR-100 and Tiny-ImageNet at $\epsilon = 1$. The performance of LA-LoRA remains stable across these configurations: the test accuracy varies within at most 0.97%. All kernel choices provide a large gain over the DP-LoRA baseline (Table 3). Among the three variants, the 5-tap kernel $G_s^{(5)}$ achieves the best overall trade-off between accuracy and efficiency, and thus is used as the default in all main experiments.

Table 19: Sensitivity of LA-LoRA to the choice of 1D binomial kernel size. Results are reported for Swin-B on CIFAR-100 and Tiny-ImageNet at $\epsilon = 1$.

| Kernel | Size | Coefficients | CIFAR-100 | Tiny-ImageNet |
|---|---|---|---|---|
| $G_s^{(3)}$ | 3 | $\frac{1}{4}[1, 2, 1]$ | $73.59_{\pm 0.60}$ | $59.92_{\pm 0.51}$ |
| $G_s^{(5)}$ | 5 | $\frac{1}{16}[1, 4, 6, 4, 1]$ | $\mathbf{74.56}_{\pm 0.52}$ | $60.68_{\pm 0.55}$ |
| $G_s^{(7)}$ | 7 | $\frac{1}{64}[1, 6, 15, 20, 15, 6, 1]$ | $73.88_{\pm 0.46}$ | $\mathbf{60.74}_{\pm 0.63}$ |

# F   DETAILED THEORETICAL ANALYSIS

## F.1   PRIVACY GUARANTEE

We report standard $(\epsilon, \delta)$-DP guarantees using the Rényi DP (RDP) accountant. Below we present the RDP definition, composition, and the conversion from RDP to $(\epsilon, \delta)$-DP. Further implementation details are available in our code and Noble et al. (2022).

**Definition 2** (Rényi DP). *For any $\lambda \in (1, \infty)$ and privacy parameter $\rho > 0$, a randomized mechanism $\mathcal{M} : \mathcal{X}^n \to \mathcal{Y}$ is said to be $(\lambda, \rho)$-RDP, if for any two neighboring datasets $\mathcal{D}$ and $\mathcal{D}'$,*

$$D_\lambda\left[\mathcal{M}(\mathcal{D}) \| \mathcal{M}(\mathcal{D}')\right] := \frac{1}{\lambda - 1} \log \mathbb{E}_{W \sim \mathcal{M}(\mathcal{D}')}\left[\left(\frac{p_{\mathcal{M}(\mathcal{D})}(W)}{p_{\mathcal{M}(\mathcal{D}')}(W)}\right)^\lambda\right] \leq \rho. \tag{17}$$

**Composition (additivity) in RDP.** RDP composes additively: if mechanisms $\mathcal{M}_1, \ldots, \mathcal{M}_S$ are applied on the same dataset (possibly adaptively), then for any fixed order $\lambda > 1$,

$$\rho_{\text{total}}(\lambda) = \sum_{s=1}^{S} \rho_s(\lambda). \tag{18}$$

**Conversion from RDP to $(\epsilon, \delta)$-DP.** If a mechanism is $(\lambda, \rho(\lambda))$-RDP for all $\lambda > 1$, then for any $\delta \in (0, 1)$ it satisfies

$$(\epsilon, \delta)\text{-DP with} \quad \epsilon(\delta) = \min_{\lambda > 1}\left\{\rho(\lambda) + \frac{\log(1/\delta)}{\lambda - 1}\right\}. \tag{19}$$

**Post-processing.** If $\mathcal{M}$ is $(\epsilon, \delta)$-DP (or $(\lambda, \rho)$-RDP) and $f$ is any (possibly randomized) mapping independent of the private data, then $f \circ \mathcal{M}$ enjoys the same privacy parameters (Dwork et al., 2014). In LA-LoRA, the Gaussian low-pass filter is such an $f$ applied to the noisy updates, which justifies the post-processing argument following Theorem 1.

## F.2   CLOSED-FORM PROJECTED GRADIENTS

We present a projection-based view to explain why the proposed local alternating update improves optimization stability compared to standard LoRA.

As discussed in Section 3, simultaneous updates of $A$ and $B$ suffer from gradient coupling (Eq. 3), amplified noise (Eq. 4), and sharper aggregated solutions. In contrast, LA-LoRA alternately updates $A$ and $B$ (Eq. 5), effectively decomposing the optimization into two sequential low-rank projections.

Let $A_k \in \mathbb{R}^{r \times n}$ and $B_k \in \mathbb{R}^{m \times r}$ denote the low-rank factors at step $k$, and $s = \alpha/r$ the LoRA scaling factor. The update to $B$ is based on solving a least-squares problem that projects the full gradient onto the column space of $A_k$:

$$\min_{\tilde{\nabla}_{B_k}\mathcal{L}} \|s(\tilde{\nabla}_{B_k}\mathcal{L})A_k - \nabla_{W_k}\mathcal{L}\|_F^2. \tag{20}$$

Here, $\|\cdot\|_F^2$ refers to the squared Frobenius norm. $\tilde{\nabla}_A\mathcal{L}$ and $\tilde{\nabla}_B\mathcal{L}$ are corresponding approximated gradients. Once the optimal direction is obtained, the update then proceeds as:

$$\begin{aligned} B_{k+1} &\leftarrow B_k - \eta\tilde{\nabla}_{B_k}\mathcal{L}, \\ W_{k+\frac{1}{2}} &\leftarrow W_k - \eta(\tilde{\nabla}_{B_k}\mathcal{L})A_k, \end{aligned} \tag{21}$$

To maintain consistency with the simultaneous update strategy, we apply the model update to an intermediate state $(k + \frac{1}{2})$. In our implementation, we treat each individual update to $A$ or $B$ as a distinct optimization step, which simplifies the analysis without ambiguity.

After performing backpropagation with respect to $B$, the gradient computed for $A$ no longer accurately reflects the full gradient at step $k$, since the model parameters have already been partially updated. Consequently, to ensure that the update to $A$ remains faithful to the full gradient, we minimize the discrepancy between the actual gradient at the intermediate model state $W_{k+\frac{1}{2}}$ and the low-rank approximation derived from $A_k$, formulated as:

$$\min_{\tilde{\nabla}_{A_k}\mathcal{L}} \|sB_{k+1}(\tilde{\nabla}_{A_k}\mathcal{L}) - \nabla_{W_{k+\frac{1}{2}}}\mathcal{L}\|_F^2. \tag{22}$$

Then, by gradient descent, we can update $A$ and the full model as

$$A_{k+1} \leftarrow A_k - \eta\tilde{\nabla}_{A_k}\mathcal{L}, \quad W_{k+1} \leftarrow W_{k+\frac{1}{2}} - \eta B_{k+1}(\tilde{\nabla}_{A_k}\mathcal{L}). \tag{23}$$

After solving the least-squares projection problems (20) and (22), we obtain their optimal solutions in closed form. Specifically, Theorem 2 states that the projected gradients can be expressed as

$$\tilde{\nabla}_{B_k}\mathcal{L} = \frac{1}{s^2}\nabla_{B_k}\mathcal{L}(A_kA_k^\top)^{-1}, \qquad \tilde{\nabla}_{A_k}\mathcal{L} = \frac{1}{s^2}(B_{k+1}^\top B_{k+1})^{-1}\nabla_{A_k}\mathcal{L},$$

where the $(A_kA_k^\top)^{-1}$ and $B_{k+1}^\top(B_{k+1})^{-1}$ terms arise from the Gram matrix inverses in the least-squares solutions, and the $\frac{1}{s^2}$ factor accounts for the LoRA scaling in both $A$ and $B$ directions.

*Proof.* For (20), differentiating the least-squares objective w.r.t. $\tilde{\nabla}_{B_k}\mathcal{L}$ and setting the derivative to zero gives

$$s^2\,\tilde{\nabla}_{B_k}\mathcal{L}\,A_kA_k^\top = s\,\nabla_{W_k}\mathcal{L}\,A_k^\top.$$

Assuming $A_kA_k^\top$ is invertible, we have

$$\tilde{\nabla}_{B_k}\mathcal{L} = \frac{1}{s}\nabla_{W_k}\mathcal{L}\,A_k^\top\,(A_kA_k^\top)^{-1}.$$

Using the LoRA gradient relation $\nabla_{B_k}\mathcal{L} = s\,\nabla_{W_k}\mathcal{L}\,A_k^\top$, we substitute:

$$\tilde{\nabla}_{B_k}\mathcal{L} = \frac{1}{s^2}\nabla_{B_k}\mathcal{L}\,(A_kA_k^\top)^{-1}.$$

The derivation for (22) is analogous, yielding

$$s^2\,B_{k+1}^\top B_{k+1}\,\tilde{\nabla}_{A_k}\mathcal{L} = s\,B_{k+1}^\top\nabla_{W_k}\mathcal{L},$$

and hence

$$\tilde{\nabla}_{A_k}\mathcal{L} = \frac{1}{s}(B_{k+1}^\top B_{k+1})^{-1}B_{k+1}^\top\nabla_{W_k}\mathcal{L} = \frac{1}{s^2}(B_{k+1}^\top B_{k+1})^{-1}\nabla_{A_k}\mathcal{L}.$$

$\square$

### F.3 STABLE FEATURE LEARNING

*Proof.* Under the regularized gradient formulation in Eq. 13, the update to the full model can be written as two half-steps.

**First half-step (updating $B$).** Using the least-squares solution for $\tilde{\nabla}_{B_k}\mathcal{L}$,

$$\begin{aligned}
W_{k+\frac{1}{2}} &= W_k - \eta\,s\,\tilde{\nabla}_{B_k}\mathcal{L}\,A_k \\
&= W_k - \eta\,s\big(\tfrac{1}{s}\nabla_{W_k}\mathcal{L}\,A_k^\top\,(A_kA_k^\top)^{-1}\big)A_k \\
&= W_k - \eta\,\nabla_{W_k}\mathcal{L}\,A_k^\top\,(A_kA_k^\top)^{-1}A_k \\
&= W_k - \eta\,\nabla_{W_k}\mathcal{L}\,Proj_{r(A_k)}. \tag{24}
\end{aligned}$$

**Second half-step (updating $A$).**    Using the least-squares solution for $\tilde{\nabla}_{A_k}\mathcal{L}$,

$$
\begin{aligned}
W_{k+1} &= W_{k+\frac{1}{2}} - \eta\, s\, B_{k+1}\, \tilde{\nabla}_{A_k}\mathcal{L} \\
&= W_{k+\frac{1}{2}} - \eta\, s\, B_{k+1}\left(\tfrac{1}{s}\,(B_{k+1}^\top B_{k+1})^{-1}B_{k+1}^\top\,\nabla_{W_{k+\frac{1}{2}}}\mathcal{L}\right) \\
&= W_{k+\frac{1}{2}} - \eta\, \underbrace{B_{k+1}(B_{k+1}^\top B_{k+1})^{-1}B_{k+1}^\top}_{Proj_{c(B_{k+1})}}\,\nabla_{W_{k+\frac{1}{2}}}\mathcal{L}.
\end{aligned}
\tag{25}
$$

Combining the two half-steps yields

$$
W_{k+1} = W_k - \eta(\nabla_{W_k}\mathcal{L})Proj_{r(A_k)} - \eta Proj_{c(B_{k+1})}(\nabla_{W_{k+\frac{1}{2}}}\mathcal{L}).
\tag{26}
$$

Here,

$$
Proj_{r(A_k)} := A_k^\top (A_k A_k^\top)^{-1} A_k, \qquad Proj_{c(B_{k+1})} := B_{k+1}(B_{k+1}^\top B_{k+1})^{-1}B_{k+1}^\top,
$$

denote the orthogonal projections onto the row space of $A_k$ (right-multiplication) and the column space of $B_{k+1}$ (left-multiplication), respectively. □

### F.4   CONVERGENCE ANALYSIS

#### F.4.1   SET UP

Following the previous work Zhang & Pilanci (2024), we provide a convergence analysis of the proposed algorithm within the over-parameterized two-layer ReLU neural network tuning problem. For a data matrix $X \in \mathbb{R}^{n \times d}$ and any arbitrary vector $u \in \mathbb{R}^d$, we consider the set of diagonal matrices $\{\mathrm{diag}([Xu \geq 0]) \mid u \in \mathbb{R}^d\}$, which take values 1 or 0 along the diagonal and indicate the possible activation patterns of the ReLU units. Let the distinct elements of this set be denoted as $D_1, \ldots, D_P$ (see Zhang & Pilanci (2024) for more details). The constant $P$ corresponds to the total number of partitions of $\mathbb{R}^d$ by hyperplanes passing through the origin that are perpendicular to the rows of $X$ Pilanci & Ergen (2020). Intuitively, $P$ can be regarded as the number of possible ReLU activation patterns associated with $X$. Pilanci & Ergen (2020) explains that a two-layer ReLU problem shares the same optimal objective with a convex problem.

$$
\min_{W_i\ i \in [P]} \frac{1}{2}\left\|\sum_{i=1}^{P} D_i X W_i - Y\right\|_F^2.
\tag{27}
$$

As we focus on fine-tuning, given a pretrained model with weights $\{W_i\}_{i=1}^P$, we perform a low-rank adaptation and express the problem in equation 27 as

$$
\min_{A_i, B_i, i=1,\cdots P} \frac{1}{2}\left\|\sum_{i=1}^{P} D_i X(W_i + B_i A_i) - Y\right\|_F^2,
\tag{28}
$$

Let $X \in \mathbb{R}^{n \times d}$, $A_i \in \mathbb{R}^{r \times c}$, $B_i \in \mathbb{R}^{d \times r}$, and $Y \in \mathbb{R}^{n \times c}$. We consider the response model

$$
Y = \sum_{i=1}^{P} D_i X\left(W_i + B_i^\star A_i^\star\right).
$$

Define

$$
X_\star := \sum_{i=1}^{P} B_i^\star A_i^\star,
$$

where the matrices $B_i^\star$ and $A_i^\star$ (and hence $X_\star$) are fixed but unknown. We use $\sigma_r(\cdot)$ to denote the $r$-th largest singular value of a matrix. Before proceeding, we first introduce the notion of the Restricted Isometry Property (RIP).

**Definition 3.** *(Restricted Isometry Property, Recht et al. (2010)) The matrix $C \in \mathbb{R}^{n \times d}$ is said to satisfy Restricted Isometry Property(RIP) with parameters $(r, \delta_r)$ if there exists constants $0 \leq \delta_r \leq 1$, for any matrices $M \in \mathbb{R}^{d \times c}$ with rank $r$, the below holds*

$$(1 - \delta_r)\|M\|_F^2 \leq \|CM\|_F^2 \leq (1 + \delta_r)\|M\|_F^2. \tag{29}$$

RIP is a widely used condition in the field of compressed sensing (Recht et al. (2010)), which states that the operator $C$ approximately preserves distances between low-rank matrices. In the absence of noise, we can establish a direct relationship between the loss function and the recovery error. If we denote $C_i := D_i X$, Problem (28) is equivalent to the problem below up to a change of labels

$$\min_{A_i, B_i, i=1,\cdots P} \mathcal{L}_c(\boldsymbol{B}, \boldsymbol{A}) := \frac{1}{2} \left\| \sum_i^P C_i (B_i A_i - X_\star) \right\|_F^2, \tag{30}$$

where $\boldsymbol{B} = \{B_1, \cdots, B_P\}$ and $\boldsymbol{A} = \{A_1, \cdots, A_P\}$.

**Notation** Inspired by the previous work Liu et al. (2025f), we introduce two local norms and their corresponding dual norms for a matrix $W \in \mathbb{R}^{k \times r}$

$$P_{A_k^i} := A_k^i (A_k^i)^\top, \quad \|W\|_{P_{A_k^i}} := \|W P_{A_k^i}^{\frac{1}{2}}\|_F, \quad \|W\|_{P_{A_k^i}^\star} := \|W P_{A_k^i}^{-\frac{1}{2}}\|_F,$$

$$P_{B_k^i} := (B_k^i)^\top B_k^i, \quad \|W\|_{P_{B_k^i}} := \|W P_{B_k^i}^{\frac{1}{2}}\|_F, \quad \|W\|_{P_{B_k^i}^\star} := \|W P_{B_k^i}^{-\frac{1}{2}}\|_F. \tag{31}$$

Here, we assume $A_k^i$ and $B_k^i$ are of full rank $r$ for any $i$. If they aren't of full rank, we can replace them with the Moore-Penrose inverse. Now we are ready to establish the convergence analysis.

### F.4.2 USEFUL LEMMA

For the $k$-th iteration, let's denote $\boldsymbol{B}_k = \{B_k^1, \cdots, B_k^P\}$ and $\boldsymbol{A}_k = \{A_k^1, \cdots, A_k^P\}$. If we apply LA-LoRA or LA-LoRA+ without momentum for Problem (30), for any $i \in [P]$, the alternating update rule as we proposed can be written as

$$A_{k+1}^i \leftarrow A_k^i - \eta (B_k^i (B_k^i)^\top)^{-1} \nabla_{A_k^i} \mathcal{L}_c(\boldsymbol{B}_k, \boldsymbol{A}_k)$$

$$B_{k+1}^i \leftarrow B_k^i - \eta \nabla_{B_k^i} \mathcal{L}_c(\boldsymbol{B}_k, \boldsymbol{A}_{k+1})((A_{k+1}^i)^\top A_{k+1}^i)^{-1}. \tag{32}$$

First, we will list some assumptions used in our analysis.

**Assumption 1.** *Suppose that $C_i = D_i X$ obeys the $r$-RIP with a constant $\delta_r$ for each $i$.*

**Assumption 2.** *Suppose that $\|C_i^T C_j\|_2 := \|X^T D_i^T D_j X\|_2 \leq \frac{1+\delta_r}{P(P-1)}$.*

Assumption 1 and 2 also adopt in Zhang & Pilanci (2024) to analyze their optimizer for LoRA. For matrix $X$ with $i.i.d$ Gaussian entries $\mathcal{N}(0, 1/d\|D_i\|_0)$, $D_i X$ satisfies RIP for a constant $\delta_r$ when $\|D_i\|_0$ is on the order of $r(d+c)/(d\delta_r^2)$. Note $\|X^\top D_i^\top D_j X\|_2 \leq \|X^\top X\|_2$ for all $(i,j)'s$. Thus bounding $\|X^\top D_i^\top D_j X\|_2$ amounts to bounding the largest singular value of the empirical covariance.

**Lemma 1.** *For a given $i \in [P]$, the gradient of Problem (30) are*

$$\nabla_{A_k^i} \mathcal{L}(\boldsymbol{B}, \boldsymbol{A}) = \sum_j^P (B_k^i)^\top (C_i)^\top C_j (B_k^j A_k^j - X_\star),$$

$$\nabla_{B_k^i} \mathcal{L}(\boldsymbol{B}, \boldsymbol{A}) = \sum_j^P (C_i)^\top C_j (B_k^j A_{k+1}^j - X_\star)(A_{k+1}^j)^\top. \tag{33}$$

*Proof.* For any given $i$ and $t$, it yields

$$\nabla_{A_k^i} \mathcal{L}(\boldsymbol{B}, \boldsymbol{A}) = \frac{\partial}{\partial A_k^i} \left\{ \frac{1}{2} \left\| \sum_j^P C_j (B_j A_j - X_\star) \right\|_F^2 \right\} = \sum_j^P (B_k^i)^\top (C_i)^\top C_j (B_k^j A_k^j - X_\star). \tag{34}$$

Similarly, we can derive the $\nabla_{B_k^i} L(\boldsymbol{B}, \boldsymbol{A})$ as shown in (33). $\qquad\square$

**Lemma 2.** *Suppose Assumption 1 and 2 holds, then we have*

$$\mathcal{L}_c(\boldsymbol{B}_k, \boldsymbol{A}_{k+1}) \leq \mathcal{L}_c(\boldsymbol{B}_k, \boldsymbol{A}_k) - c_1 \max_i \left\|\nabla_{A_k^i} \mathcal{L}_c(\boldsymbol{B}_k, \boldsymbol{A}_k)\right\|^2_{P^\star_{B_k^i}},$$

$$\mathcal{L}_c(\boldsymbol{B}_{k+1}, \boldsymbol{A}_{k+1}) \leq \mathcal{L}_c(\boldsymbol{B}_k, \boldsymbol{A}_{k+1}) - c_1 \max_i \left\|\nabla_{B_k^i} \mathcal{L}_c(\boldsymbol{B}_k, \boldsymbol{A}_{k+1})\right\|^2_{P^\star_{A_{k+1}^i}},$$

$$(35)$$

*where* $c_1 = P(\eta - \frac{\eta^2(1+\delta_r+\frac{1}{P})}{2})$.

*Proof.* Using the update rule in (32), we have

$$\begin{aligned}
\mathcal{L}_c(\boldsymbol{B}_k, \boldsymbol{A}_{k+1}) &= \frac{1}{2}\left\|\sum_i^P C_i(B_k^i A_{k+1}^i - X_\star)\right\|^2_F \\
&= \frac{1}{2}\left\|\sum_i^P C_i\left(B_k^i\left(A_k^i - \eta((B_k^i)^\top B_k^i)^{-1}\nabla_{A_k^i}\mathcal{L}_c(\boldsymbol{B}_k, \boldsymbol{A}_k)\right) - X_\star\right)\right\|^2_F \\
&= \frac{1}{2}\left\|\sum_i^P C_i(B_k^i A_k^i - X_\star)\right\|^2_F \\
&\quad + \underbrace{\frac{\eta^2}{2}\left\|\sum_i^P C_i B_k^i((B_k^i)^\top B_k^i)^{-1}\nabla_{A_k^i}\mathcal{L}_c(\boldsymbol{B}_k, \boldsymbol{A}_k)\right\|^2_2}_{T_1} \\
&\quad - \underbrace{\eta\left\langle\sum_i^P C_i(B_k^i A_k^i - X_\star), \sum_i^P C_i B_k^i((B_k^i)^\top B_k^i)^{-1}\nabla_{A_k^i}\mathcal{L}_c(\boldsymbol{B}_k, \boldsymbol{A}_k)\right\rangle}_{T_2}
\end{aligned}$$

$$(36)$$

For $T_1$, recalling Lemma 1, then we have

$$\begin{aligned}
T_1 &\leq \frac{\eta^2}{2}\sum_i^P \|C_i B_k^i((B_k^i)^\top B_k^i)^{-1}\nabla_{A_k^i}\mathcal{L}_c(\boldsymbol{B}_k, \boldsymbol{A}_k)\|_F^2 \\
&\quad + \frac{\eta^2}{2}\sum_{i\neq j}\left\langle C_i B_k^i((B_k^i)^\top B_k^i)^{-1}\nabla_{A_k^i}\mathcal{L}_c(\boldsymbol{B}_k, \boldsymbol{A}_k), C_j B_k^j((B_k^j)^\top B_k^j)^{-1}\nabla_{A_k^j}\mathcal{L}_c(\boldsymbol{B}_k, \boldsymbol{A}_k)\right\rangle \\
&\overset{(a)}{\leq} \frac{\eta^2(1+\delta_r)}{2}P\max_i\|\nabla_{A_k^i}\mathcal{L}_c(\boldsymbol{B}_k, \boldsymbol{A}_k)\|^2_{P^\star_{B_k^i}} \\
&\quad + \frac{\eta^2}{2}\max_{i\neq j}\|C_i^\top Cj\|_2 P(P-1)\max_i\|\nabla_{A_k^i}\mathcal{L}_c(\boldsymbol{B}_k, \boldsymbol{A}_k)\|^2_{P_{B_k^i}} \\
&\overset{(b)}{\leq} \frac{\eta^2(1+\delta_r+\frac{1}{P})}{2}P\max_i\|\nabla_{A_k^i}\mathcal{L}_c(\boldsymbol{B}_k, \boldsymbol{A}_k)\|^2_{P_{B_k^i}},
\end{aligned}$$

$$(37)$$

where (a) uses Cauchy Inequality, Assumption 1 and the fact that $\|B_k^i((B_k^i)^\top B_k^i)^{-\frac{1}{2}}\|_2^2 = 1$, (b) uses the assumption that $\max_{i\neq j}\|C_j^\top C_j\|_2 \leq \frac{(1+\delta_r)}{P(P-1)}$.

For $T_2$, using Lemma 1 again, we have

$$
\begin{aligned}
T_2 &= \eta \left\langle \sum_j^P C_j(B_k^j A_k^j - X_\star), \sum_j^P C_j B_k^j((B_k^j)^\top B_k^j)^{-1}\nabla_{A_k^j}\mathcal{L}_c(\boldsymbol{B}_k, \boldsymbol{A}_k) \right\rangle \\
&= \eta \sum_j^P \left\langle \sum_i^P C_i(B_k^i A_k^i - X_\star), C_j B_k^j((B_k^j)^\top B_k^j)^{-1}\nabla_{A_k^j}\mathcal{L}_c(\boldsymbol{B}_k, \boldsymbol{A}_k) \right\rangle \\
&= \eta \sum_i^P \left\| \nabla_{A_k^i}\mathcal{L}_c(\boldsymbol{B}_k, \boldsymbol{A}_k) \right\|_{P_{B_k^i}^\star}^2 \\
&\leq \eta P \max_i \| \nabla_{A_k^i}\mathcal{L}_c(\boldsymbol{B}_k, \boldsymbol{A}_k) \|_{P_{B_k^i}}^2 .
\end{aligned}
\tag{38}
$$

To sum up, it yields

$$
\mathcal{L}_c(\boldsymbol{B}_k, \boldsymbol{A}_{k+1}) \leq \mathcal{L}_c(\boldsymbol{B}_k, \boldsymbol{A}_k) - \left( \eta - \frac{\eta^2(1 + \delta_r + \frac{1}{P})}{2} \right) P \max_i \left\| \nabla_{A_k^i}\mathcal{L}_c(\boldsymbol{B}_k, \boldsymbol{A}_k) \right\|_{P_{B_k^i}^\star}^2 . \tag{39}
$$

Similarly, we can induce

$$
\mathcal{L}_c(\boldsymbol{B}_{k+1}, \boldsymbol{A}_{k+1}) \leq \mathcal{L}_c(\boldsymbol{B}_k, \boldsymbol{A}_{k+1}) - \left( \eta - \frac{\eta^2(1 + \delta_r + \frac{1}{P})}{2} \right) P \max_i \left\| \nabla_{B_k^i}\mathcal{L}_c(\boldsymbol{B}_k, \boldsymbol{A}_{k+1}) \right\|_{P_{A_{k+1}^i}^\star}^2 . \tag{40}
$$

$\square$

**Lemma 3.** *Suppose Assumption 1 holds, then, for any $i \in [P]$, we have*

$$
\begin{aligned}
\|\nabla_{A_k^i}\mathcal{L}_c(\boldsymbol{B}_k, \boldsymbol{A}_k)\|_{P_{B_k^i}^\star}^2 &\geq 2(1 - \delta_r)\mathcal{L}_c(\boldsymbol{B}_k, \boldsymbol{A}_k), \\
\|\nabla_{B_k^i}\mathcal{L}_c(\boldsymbol{B}_t, \boldsymbol{A}_{k+1})\|_{P_{A_{k+1}^i}^\star}^2 &\geq 2(1 - \delta_r)\mathcal{L}_c(\boldsymbol{B}_k, \boldsymbol{A}_{k+1}).
\end{aligned}
\tag{41}
$$

*Proof.* See Lemma 6 in Liu et al. (2025f) for the detailed proof. $\square$

**Theorem 5.** *Assume for any $i \in [p]$ the matrix $C_i = D_i X$ satisfies the rank $r$-RIP with constant $\delta_r$ (Assumption 1) and $0 \leq \eta \leq \frac{1}{1+\delta_r+\frac{1}{P}}$, then LA-LoRA without momentum solves the over-parameterized problem leads to*

$$
\mathcal{L}_c(\boldsymbol{B}_{k+1}, \boldsymbol{A}_{k+1}) \leq (1 - \eta_c)^2 \mathcal{L}_c(\boldsymbol{B}_k, \boldsymbol{A}_k), \tag{42}
$$

*and*

$$
\left\| \sum_i^P B_k^i A_k^i - X_\star \right\|_F^2 \leq \frac{1 + \delta_r}{1 - \delta_r}(1 - \eta_c)^{2k} \left\| \sum_i^P B_0^i A_0^i - X_\star \right\|_F^2, \tag{43}
$$

*where $\eta_c = 2P(1 - \delta_r)\left( \eta - \frac{\eta^2(1+\delta_r+\frac{1}{P})}{2} \right)$.*

*Proof.*

$$
\begin{aligned}
\mathcal{L}_c(\boldsymbol{B}_{k+1}, \boldsymbol{A}_{k+1}) &\leq \mathcal{L}_c(\boldsymbol{B}_k, \boldsymbol{A}_{k+1}) - \left( \eta - \frac{\eta^2(1 + \delta_r + \frac{1}{P})}{2} \right) P \max_i \left\| \nabla_{B_k^i}\mathcal{L}_c(\boldsymbol{B}_k, \boldsymbol{A}_{k+1}) \right\|_{P_{A_{k+1}^i}^\star}^2 \\
&\leq \mathcal{L}_c(\boldsymbol{B}_k, \boldsymbol{A}_{k+1}) - \left( \eta - \frac{\eta^2(1 + \delta_r + \frac{1}{P})}{2} \right) 2P(1 - \delta_r)\mathcal{L}_c(\boldsymbol{B}_k, \boldsymbol{A}_{k+1}) \\
&\leq \left( 1 - 2P(1 - \delta_r)\left( \eta - \frac{\eta^2(1 + \delta_r + \frac{1}{P})}{2} \right) \right) \mathcal{L}_c(\boldsymbol{B}_k, \boldsymbol{A}_{k+1}) \\
&\leq (1 - \eta_c)^2 \mathcal{L}_c(\boldsymbol{B}_k, \boldsymbol{A}_k),
\end{aligned}
\tag{44}
$$

where we apply Lemma 2 and 3 and $\eta_c = 2P(1-\delta_r)\left(\eta - \frac{\eta^2(1+\delta_r+\frac{1}{P})}{2}\right)$. Moreover, under Assumption 1, we have

$$\left\|\sum_i^P B_k^i A_k^i - X_\star\right\|_F^2 \leq \frac{1+\delta_r}{1-\delta_r}(1-\eta_c)^{2k}\left\|\sum_i^P B_0^i A_0^i - X_\star\right\|_F^2. \tag{45}$$

$\square$

### F.5 COMPARISON WITH FFA-LoRA AND RoLoRA

**Corollary 1** (Comparison with FFA-LoRA and RoLoRA)**.** *Under the same setting as Theorem 3, consider the following two baselines.*

**FFA-LoRA (frozen** $A$**).** *Let $A_k \equiv A_0$ and define the fixed row-space subspace $\mathcal{S}_0 = \{\Delta W \in \mathbb{R}^{m\times n} : \mathrm{row}(\Delta W) \subseteq r(A_0)\}$. Then $W_k - W_0 \in \mathcal{S}_0$ for all $k$. Hence, for any target rank-r solution $W^\star$ with $W^\star - W_0 \notin \mathcal{S}_0$, $\inf_k \|W_k - W^\star\|_F \geq \mathrm{dist}(W^\star - W_0, \mathcal{S}_0) > 0$.*

**RoLoRA (round-wise alternation).** *For one round updating $B$ with $A$ fixed, followed by one round updating $A$ with $B$ fixed, we show*

$$W_{k+2} = W_k - \eta\left(\nabla_{W_k}\mathcal{L}\right)Proj_{r(A_{k+1})} - \eta\,Proj_{c(B_{k+1})}\left(\nabla_{W_{k+\frac{1}{2}}}\mathcal{L}\right) + \mathcal{E}_k,$$

*where the first two terms coincide with the LA-LoRA update in Eq. (26) and the stale-block remainder satisfies $\|\mathcal{E}_k\|_F = \mathcal{O}(\eta^2\|\nabla_{W_k}\mathcal{L}\|_F)$.*

*In contrast, LA-LoRA alternates $B$ and $A$ within each local iteration and exactly matches Eq. (26), i.e., it imposes no fixed-row-space constraint as in FFA-LoRA and has $\mathcal{E}_k = 0$.*

Next, We provide the details underlying Corollary 1.

**FFA-LoRA: fixed row-space subspace.** FFA-LoRA freezes $A$ at some initialization $A_0$ and only updates $B$. At iteration $k$ we can write

$$W_k = W_0 + sB_kA_0,$$

so

$$W_k - W_0 \in \mathcal{S}_0 \quad \text{with} \quad \mathcal{S}_0 = \{\Delta W \in \mathbb{R}^{m\times n} : \mathrm{row}(\Delta W) \subseteq r(A_0)\}.$$

Let $W^\star$ be a target (e.g., optimal) rank-$r$ solution. If $W^\star - W_0 \notin \mathcal{S}_0$, then by projection geometry

$$\inf_k \|W_k - W^\star\|_F \geq \inf_{\Delta W \in \mathcal{S}_0} \|\Delta W - (W^\star - W_0)\|_F = \mathrm{dist}(W^\star - W_0, \mathcal{S}_0) > 0,$$

which is exactly the irreducible approximation gap stated for FFA-LoRA.

**RoLoRA: stale-block remainder $\mathcal{E}_k$.** RoLoRA alternates between updating $B$ and $A$ at the level of communication rounds. Consider two consecutive local steps forming one full "$B$-then-$A$" cycle. Let $W_{k+1/2}$ denote the intermediate weight after updating $B$ but before updating $A$. Using the projected-gradient view of Theorem 2, we have

$$W_{k+1/2} = W_k - \eta\,Proj_{c(B_{k+1})}\left(\nabla_{W_k}\mathcal{L}\right), W_{k+2} = W_{k+1/2} - \eta\left(\nabla_{W_{k+1/2}}\mathcal{L}\right)Proj_{r(A_{k+1})}.$$

Adding these relations gives

$$W_{k+2} = W_k - \eta\,Proj_{c(B_{k+1})}\left(\nabla_{W_k}\mathcal{L}\right) - \eta\left(\nabla_{W_{k+1/2}}\mathcal{L}\right)Proj_{r(A_{k+1})}.$$

We add and subtract the LA-LoRA update in Eq. (26) and define the remainder

$$\mathcal{E}_k = -\eta\Big(Proj_{c(B_{k+1})}\big(\nabla_{W_k}\mathcal{L} - \nabla_{W_{k+\frac{1}{2}}}\mathcal{L}\big) + \big(\nabla_{W_{k+1/2}}\mathcal{L} - \nabla_{W_k}\mathcal{L}\big)Proj_{r(A_{k+1})}\Big),$$

which yields the decomposition

$$W_{k+2} = W_k - \eta\left(\nabla_{W_k}\mathcal{L}\right)Proj_{r(A_{k+1})} - \eta\,Proj_{c(B_{k+1})}\big(\nabla_{W_{k+\frac{1}{2}}}\mathcal{L}\big) + \mathcal{E}_k.$$

Assuming that $\mathcal{L}$ is $L$-smooth, we have

$$\|\nabla_{W_{k+1/2}}\mathcal{L} - \nabla_{W_k}\mathcal{L}\|_F \leq \mathcal{L}\|W_{k+1/2} - W_k\|_F.$$

From the first update, $W_{k+1/2} - W_k = -\eta\, Proj_{c(B_{k+1})}(\nabla_{W_k}\mathcal{L})$, and the projector has operator norm at most 1, so $\|W_{k+1/2} - W_k\|_F \leq \eta\|\nabla_{W_k}\mathcal{L}\|_F$. Using again that the projectors have norm at most 1, we obtain

$$\begin{aligned}
\|\mathcal{E}_k\|_F &\leq 2\eta\,\|\nabla_{W_{k+1/2}}\mathcal{L} - \nabla_{W_k}\mathcal{L}\|_F \\
&\leq 2\eta\mathcal{L}\|W_{k+1/2} - W_k\|_F \\
&\leq 2\eta^2\mathcal{L}\|\nabla_{W_k}\mathcal{L}\|_F.
\end{aligned} \tag{46}$$

Under the infinite-width scaling used in Theorem 3, the Frobenius norm $\|\nabla_{W_k}\mathcal{L}\|_F$ itself scales with width, so the effective contribution of $\mathcal{E}_k$ to the feature dynamics is of order $\mathcal{O}(\eta^2\|\nabla_{W_k}\mathcal{L}\|_F)$ when $\eta = \mathcal{O}(1)$.

## G    TABLE OF NOTATIONS

Table 20 summarizes the main notations used throughout the paper.

Table 20: Summary of the main notations.

| Notation | Description |
|---|---|
| $W_0 \in \mathbb{R}^{m \times n}$ | Pre-trained backbone weight matrix |
| $A, B$ | LoRA down-projection/up-projection matrices |
| $r$ | LoRA rank, $r \ll \min\{m, n\}$ |
| $\alpha$ | LoRA scaling hyperparameter |
| $s = \alpha/r$ | LoRA scaling factor |
| $\nabla_A\mathcal{L}$, $\nabla_B\mathcal{L}$ | Gradients of loss w.r.t. $A$ and $B$ |
| $\nabla_W\mathcal{L}$ | Gradient of loss w.r.t. full weight $W$ |
| $\mathcal{L}(\cdot)$ | Local loss function |
| $\mathcal{D}_i$ | Local dataset of client $i$ |
| $N$ | Total number of clients |
| $q$ | Client sampling rate per round |
| $\mathcal{C}_t$ | Set of selected clients in round $t$ |
| $T$ | Number of communication rounds |
| $K$ | Number of local update steps per round |
| $b$ | Local data sampling rate |
| $R = |\mathcal{D}_i|$ | Size of local dataset on client $i$ |
| $\mathcal{B}_i$ | Mini-batch sampled from $\mathcal{D}_i$ |
| $C$ | Per-sample $\ell_2$ clipping norm |
| $\sigma$ | Gaussian noise multiplier for DP |
| $g_{ij}$ | Clipped gradient for sample $j$ on client $i$ |
| $g_i$ | Noisy averaged gradient on client $i$ |
| $\mathcal{N}(0, C^2\sigma^2)$ | Gaussian DP noise with variance $C^2\sigma^2$ |
| $\widehat{g}_i$ | Smoothed gradient after applying $G_s$ |
| $\epsilon, \delta$ | $(\epsilon, \delta)$-differential privacy parameters |
| $X_\star$ | Target low-rank matrix in theory |
| $Proj_{r(A)}, Proj_{c(B)}$ | Projection onto row space of $A$, Projection onto column space of $B$ |
| $H, \lambda_{\max}(H)$ | Hessian of the loss and its maximum eigenvalue (sharpness) |
| $\delta_r$ | Rank-$r$ RIP constant of matrices $C_i$ |

## H    LLM USAGE

Large Language Models (LLMs) were used to aid in the writing and polishing of the manuscript. Specifically, we used an LLM to assist in refining the language, improving readability, and ensuring

clarity in various sections of the paper. The model helped with tasks such as sentence rephrasing, grammar checking, and enhancing the overall flow of the text.

It is important to note that the LLM was not involved in the ideation, research methodology, or experimental design. All research concepts, ideas, and analyses were developed and conducted by the authors. The contributions of the LLM were solely focused on improving the linguistic quality of the paper, with no involvement in the scientific content or data analysis.

