# OpenReview forum: "Rethinking LoRA for Privacy-Preserving Federated Learning in Large Models"
_ICLR.cc/2026/Conference — ICLR 2026 Poster_

### Official Review · Reviewer_cWrF · 2025-10-23

**Soundness:** 3
**Presentation:** 3
**Contribution:** 2
**Rating:** 6
**Confidence:** 3

**Summary:**

This paper addresses three key challenges in federated DP-LoRA: gradient coupling, noise amplification, and loss sharpness. The authors propose LA-LORA, which employs two components: a local alternating update strategy to tackle gradient coupling and noise amplification, and a low-pass smoothing filter applied pre-aggregation to mitigate solution sharpness. Theoretically, the paper's analysis demonstrates that this alternating update avoids a destabilizing "across term" inherent in standard joint updates, leading to more stable optimization. Experimentally, LA-LORA outperforms existing methods like DP-LORA and RoLoRA on both vision and language benchmarks.

**Strengths:**

The paper's presentation is clear and the writing flow is well-organized. Also a clear reproducibility statement is supported with codes.

**Weaknesses:**

1. The theoretical analysis convincingly argues for alternating updates vs. joint updates but fails to differentiate LA-LORA from other alternating baselines such as FFA-LORA, RoLoRA.
2. The analysis lacks a deep discussion on the relative importance of the three identified challenges. It remains unclear which challenge contributes most to performance degradation or how their respective impacts differ across vision versus language tasks.
3. The paper re-frames the aggregation challenge as "sharpness" but fails to address the underlying mathematical "aggregation bias"( $\mathbb{E}[\bar{B}]\mathbb{E}[\bar{A}] \neq \mathbb{E}[\overline{BA}]$), and it is unclear how the low-pass filter resolves this fundamental discrepancy.

**Questions:**

1. What explains LA-LORA's empirical gains over FFA-LORA and RoLoRA?
2. Why is LA-LORA's performance gain dramatically larger in vision (+16.83% on Swin-B) than in language (+2.48% on QNLI)?
3. How does the algorithm perform (the final test accuracy) against other baselines in the non-private setting?
4. How sensitive is LA-LORA to the choice of local steps K?

---

> ### Author Response · Authors · 2025-11-25
> **Response (1/3)**
>
> We sincerely thank you for the insightful comments.
>
> > **Weakness 1.** The theoretical analysis convincingly argues for alternating updates vs. joint updates but fails to differentiate LA-LORA from other alternating baselines such as FFA-LORA, RoLoRA.
>
> **Response for Weakness 1:**
> In **Appendix E.5** of the revision, we clearly distinguish the theoretical insights of LA-LoRA from FFA-LoRA and RoLoRA.
>
> **(1) FFA-LoRA (frozen \(A\))**
>    Freezing $A\_k \equiv A\_0$ implies
>    $W\_k = W\_0 + s B\_k A\_0,W\_k - W\_0 \in \mathcal{S}\_0 := \{\Delta W \in \mathbb{R}^{m\times n} : \mathrm{row}(\Delta W)\subseteq r(A\_0)\}.$
>    Thus FFA-LoRA optimizes in a fixed row-space subspace. For any target rank-\(r\) solution $W^\star$ with $W^\star - W\_0 \notin \mathcal{S}\_0$,
>    $
>    \inf\_k \\|W\_k - W^\star\\|\_F \\ge\ \mathrm{dist}(W^\star - W\_0,\mathcal{S}\_0) \> 0,
>    $
>    i.e., it has an irreducible approximation bias. In contrast, LA-LoRA updates both $A\_k$ and $B\_k$ and can represent any rank-$r$ update, so it does not suffer from this constraint.
>
> **(2) RoLoRA (round-wise alternation)**
>    For one round updating $A$ with $A$ fixed, followed by one round updating $A$ with $B$ fixed, we show
>    $
>    W\_{k+2}
>    = W\_k
>      - \eta(\nabla\_{W\_k}\mathcal{L})\mathrm{Proj}\_{r(A\_{k+1})}
>      - \eta\mathrm{Proj}\_{c(B\_{k+1})}(\nabla\_{W\_{k+\frac12}}\mathcal{L})
>      + \mathcal{E}\_k,
>    $
>    where the first two terms coincide with the LA-LoRA update in Eq. (24) and the stale-block remainder satisfies
>    $
>    \\|\mathcal{E}\_k\\|\_F = \mathcal{O}\bigl(\eta^2\\|\nabla\_{W\_k}\mathcal{L}\\|\_F\bigr).
>    $
>    Under the same infinite-width scaling and $\eta = \mathcal{O}(1)$, $\mathcal{E}\_k$ is of the same order as the projected terms, while LA-LoRA alternates within each local iteration and has $\mathcal{E}\_k = 0$.
>
>
>
> > **Weakness 2.** The analysis lacks a deep discussion on the relative importance of the three identified challenges. It remains unclear which challenge contributes most to performance degradation or how their respective impacts differ across vision versus language tasks.
>
> **Response for Weakness 2:**
> We add an ablation subsection in the **Appendix C.3** together with **Table 16**, which decomposes the roles of **noise amplification**, **gradient coupling**, and **loss sharpness** for both vision and language.
>
> **Table 16. Ablation of LA-LoRA components and the three challenges (noise amplification,
> gradient coupling, and loss sharpness) on GLUE ($\beta = 0.3$) and image classification
> ($\beta = 0.1$), $\epsilon = 1$. We report test accuracy (\%) and the maximum Hessian
> eigenvalue $\lambda_{\max}(H)$ on LoRA parameters.**
> | Method           | QQP (RoBERTa-Base)} |  $\lambda_{\max}(H)$ | CIFAR-100 (Swin-B)} |  $\lambda_{\max}(H)$              |
> |------------------|------------------------|------------------------|------------------------|------------------------|
> | DP-LoRA          | 84.02                    | 43.74                  | 55.98                  | 101.62                 |
> | DP-LoRA(+filter) | 85.63                     | 41.36 (↓ 2.38)         | 67.95                  | 82.33 (↓ 21.29)        |
> | LA-LoRA(-filter) | 85.95                   | 40.82 (↓ 2.92)         | 69.87                  | 64.77 (↓ 36.85)        |
> | **LA-LoRA**      | **86.41**                 | 40.22 (↓ 3.52)         | **74.56**              | 55.76 (↓ 45.86)        |
>
> - **Noise amplification.**
>   Comparing **DP-LoRA** and **DP-LoRA(+filter)** isolates the effect of the low-pass filter. On Swin-B/CIFAR-100, accuracy improves from 55.98\% to 67.95\%, and $\lambda_{\max}(H)$ drops from $101.62$ to $80.33$, showing that suppressing noise amplification alone already yields substantial gains and a smoother loss landscape.
>
> - **Gradient coupling.**
>   Comparing **DP-LoRA** and **LA-LoRA(-filter)** isolates the effect of locally alternating $B$ and $A$ at the same noise level. On Swin-B/CIFAR-100, accuracy improves from 55.98\% to 69.87\% and $\lambda_{\max}(H)$ decreases further to 64.77, indicating that mitigating gradient coupling is even more critical for deep vision backbones and strongly reduces loss sharpness.
>
> - **Joint effect and sharpness.**
>
>   **LA-LoRA** combines both components and achieves the best accuracy and the smallest $\lambda_{\max}(H)$ on both QQP and CIFAR-100. Although sharpness is not controlled by an explicit algorithmic parameter, the decrease of $\lambda_{\max}(H)$ across these methods indicates that progressively addressing noise amplification and gradient coupling systematically flattens the loss landscape.

---

> > ### Author Response · Authors · 2025-11-25
> > **Response (2/3)**
> >
> > Thank you for your detailed feedback.
> > > **Weakness 3.** The paper re-frames the aggregation challenge as "sharpness" but fails to address the underlying mathematical "aggregation bias"$(\mathbb{E}[\bar B] \mathbb{E}[\bar A] \neq \mathbb{E}[\overline{BA}]\)$, and it is unclear how the low-pass filter resolves this fundamental discrepancy.
> >
> > **Response to Weakness 3.**
> > The “aggregation bias” can be written. Let client LoRA factors be
> > $\(B\_i, A\_i\)$ and define
> > $\bar B = \mathbb{E}[B\_i]$, $\bar A = \mathbb{E}[A\_i]$.
> > Write
> > $
> > B\_i = \bar B + \Delta B\_i,
> > A\_i = \bar A + \Delta A\_i,
> > \mathbb{E}[\Delta B\_i] = \mathbb{E}[\Delta A\_i] = 0.
> > $
> > Then
> > $
> > \mathbb{E}[B\_i A\_i] - \bar B \bar A
> > = \mathbb{E}[\Delta B\_i \Delta A\_i],
> > $
> > and hence, by Cauchy–Schwarz,
> > $
> > \bigl\\|\mathbb{E}[B\_i A\_i] - \bar B \bar A\bigr\\|\_F
> > \le
> > \Bigl(\mathbb{E}\\|\Delta B\_i\\|\_F^2\Bigr)^{\frac12}
> > \Bigl(\mathbb{E}\\|\Delta A\_i\\|\_F^2\Bigr)^{\frac12}.
> > $
> > Thus the aggregation bias is entirely controlled by the variance and covariance of the client deviations $\Delta B\_i, \Delta A\_i$.
> > In this formulation, the two components of LA-LoRA act as follows:
> >
> > - the **low-pass filter** reduces $\mathbb{E}\\|\Delta B\_i\\|\_F^2$ and
> >   $\mathbb{E}\\|\Delta A\_i\\|\_F^2$ (DP noise and client drift are smoothed before aggregation);
> >
> > - the **locally alternating updates** align $B\_i$ and $A_i$ with the same projected gradient,
> >   reducing the covariance term $\mathbb{E}[\Delta B\_i \Delta A\_i]$.
> >
> > Empirically, the decrease of the LoRA Hessian eigenvalue $\lambda_{\max}(H)$ from DP-LoRA to LA-LoRA in **Table 15** is consistent with a reduction of this aggregation bias in the low-rank subspace.
> >
> >
> > > **Question 1.** What explains LA-LORA's empirical gains over FFA-LORA and RoLoRA?
> >
> > **Response for Question 1:**
> > In DP (and DPFL) training, some accuracy loss is unavoidable. Our goal is therefore not to eliminate this loss, but to improve the privacy–utility trade-off by making DP LoRA fine-tuning more effective.
> >
> > Concretely, LA-LoRA offers two main advantages under the same privacy budget:
> > **(1) Higher final accuracy.** LA-LoRA consistently achieves higher test accuracy than FFA-LoRA and RoLoRA across both vision and language benchmarks. For example, when fine tuning Swin B with $\epsilon =1$, LA-LoRA reaches 74.56\% on CIFAR-100 which is **12.62\%** higher than FFA-LoRA and **6.68\%** higher than RoLoRA, and 60.68\% on Tiny-ImageNet which is **21.35\%** and **16.83\%** higher than FFA-LoRA and RoLoRA respectively (see Table 3).
> > **(2) Faster convergence.** As shown in **Figure 6**, under $\epsilon =1$ on CIFAR-100 and Tiny-ImageNet, the LA-LoRA curve rises much faster and stays above the curves of DP-LoRA, FFA-LoRA and RoLoRA across communication rounds, which means that for any fixed target accuracy LA-LoRA reaches it in fewer communication rounds and shows smoother and more stable training under DP noise.
> >
> >
> > > **Question 2.** Why is LA-LORA's performance gain dramatically larger in vision (+16.83% on Swin-B) than in language (+2.48% on QNLI)?
> >
> > **Response for Question 2:**
> > **The main reason is the difference in model, task difficulty.** Compared with our language setup, which fine tunes a strong pretrained encoder on QNLI as a relatively simple binary classification task, the vision setting uses Swin-B for multi class image classification on CIFAR-100 and Tiny-ImageNet with hundreds of classes and much lower non private baseline accuracy. The vision transformer also has higher dimensional representations and gradients, which makes it more sensitive to gradient clipping and injected DP noise. In this regime DP training removes a larger portion of useful signal from the vision gradients, and by reducing gradient noise amplification LA-LoRA can recover a much larger fraction of the lost accuracy on Swin-B, leading to the much larger gains observed in vision.
> >
> > In addition, to our knowledge, prior FFA-LoRA and RoLoRA have been evaluated only on language models. Our work is the first to systematically study LoRA fine tuning for vision transformers in DPFL, where the tasks are harder and DP noise causes a much larger degradation. This further explains why the performance gains of LA-LoRA are particularly pronounced on vision benchmarks.

---

> ### Author Response · Authors · 2025-11-25
> **Response (3/3)**
>
> We appreciate your insightful suggestions.
> > **Question 3.** How does the algorithm perform (the final test accuracy) against other baselines in the non-private setting?
>
> **Response for Question 3:**
> We report non-private experiments in Appendix A.7 (Tables 10 and 11) and Appendix B.5 (Table 13).
>
> **(1) Vision (Appendix A.7, Table 11)**.
> Table 11 reports non-private fine tuning of Swin-B on CIFAR-100 and Tiny ImageNet. In this setting LA-LoRA achieves final test accuracy that is on par with or slightly higher than standard LoRA and the other alternating variants. This shows that our update scheme does not hurt performance in the non private regime and that the large gains reported in the main tables come from better handling of DP noise rather than from relaxing the task or the model.
>
> **Table 11: Non-private federated training for Swin-B on CIFAR-100 and Tiny-ImageNet.**
> | Method     | CIFAR-100 | Tiny-ImageNet |
> |-----------|-----------|---------------|
> | DP-LoRA   | 83.21     | 81.37         |
> | FFA-LoRA  | 81.65     | 80.76         |
> | RoLoRA    | 83.25     | 81.03         |
> | **LA-LoRA** | **84.21** | **82.58**   |
>
>
> **(2) Language (Appendix B.5, Table 13)**.
> Table 13 reports the corresponding non-private and $\epsilon = 1$ results on QNLI and the other language tasks under Dirichlet $\beta = 0.3$. LA-LoRA matches or slightly improves over the other baselines, and all methods are closer to their non private upper bound than in the vision case, which is consistent with our discussion in Question 2. Overall these results confirm that LA-LoRA remains competitive without privacy and that its main benefit is to narrow the gap between DP training and the non-private performance.
>
> **Table 13: Test accuracy(%) of RoBERTa-Base on SST-2, QNLI, QQP and MNLI, Dirichlet $\beta = 0.3$.**
>
> | Privacy     | Method   | SST-2 | QNLI  | QQP   | MNLI  | Avg.  |
> |------------|----------|-------|-------|-------|-------|-------|
> | Non-private| DP-LoRA  | 92.07 | 86.25 | 84.02 | 81.22 | 85.89 |
> |            | FFA-LoRA | 92.56 | 87.53 | 85.36 | 81.54 | 86.75 |
> |            | RoLoRA   | 93.65 | 88.65 | 85.52 | 82.22 | 87.51 |
> |            | **LA-LoRA** | **93.94** | **89.96** | **86.41** | **83.32** | **88.41** |
> | $\epsilon = 1$      | DP-LoRA  | 90.13 | 83.79 | 83.28 | 79.80 | 84.25 |
> |            | FFA-LoRA | 90.62 | 84.63 | 84.10 | 80.89 | 85.08 |
> |            | RoLoRA   | 91.74 | 85.86 | 84.21 | 81.35 | 85.78 |
> |            | **LA-LoRA** | **92.11** | **87.13** | **85.04** | **82.27** | **86.64** |
>
>
> > **Question 4.** How sensitive is LA-LORA to the choice of local steps K?
>
> **Response for Question 4:**
> We add a sensitivity study of the local steps K in **Appendix C.4 (Table 17)**. In this experiment we keep the clipping norm, noise multiplier, and the number of communication rounds fixed, and fine-tune Swin-B on CIFAR-100 and Tiny-ImageNet.
>
> **Table 17: LA-LoRA test accuracy (%) with different local steps K (Swin-B).**
>
> | Dataset         | $\sigma$ | K = 10 | K = 20 | K = 30 | K = 50 |
> |----------------|-----------|--------|--------|--------|--------|
> | CIFAR-100 (%)  | 0.56      | 72.13  | 74.56  | 75.21  | 75.26  |
> | Tiny-ImageNet (%) | 0.283  | 58.54  | 60.68  | 61.24  | 61.32  |
>
>
> As shown in Table 17, increasing K from 10 to 50 changes the final test accuracy by about 3\%. Larger K generally yields slightly better accuracy but also requires more local computation per communication round and, under standard DP composition, leads to a larger privacy budget $\epsilon$ when the noise multiplier is fixed. To balance model quality, privacy, and client cost, we therefore use K = 20 as the default setting in all main experiments.

---

> > ### Author Response · Authors · 2025-11-28
> > **Thank you very much！Reviewer cWrF**
> >
> > We sincerely thank you for your follow-up and your thorough review. We hope that we have addressed the comments in a satisfactory manner.

---

### Official Review · Reviewer_gf7V · 2025-10-26

**Soundness:** 2
**Presentation:** 3
**Contribution:** 2
**Rating:** 6
**Confidence:** 3

**Summary:**

This paper proposes LA-LoRA (Local Alternating LoRA), a new variant of LoRA for fine-tuning models under differentially private federated learning (DPFL). The authors identify three key challenges in applying conventional Low-Rank Adaptation (LoRA) to DPFL: 1)gradient coupling 2) noise amplification, and 3) sharpness of aggregated solutions, which degrade model performance under privacy constraints. To address these, LA-LoRA introduces a local alternating update strategy, where LoRA’s two low-rank matrices are updated sequentially rather than simultaneously with a low-pass filter. Some experimental results are presented to demonstrate the effectiveness of the proposed algorithm, compared to DP-LoRA, FFA-LoRA and RoLoRA.

**Strengths:**

1) The paper presents clear problem identification and the potential reasons of the problems.
2) The proposed algorithm is elegant, clear and effective.
3) Some analysis about the stability is presented to demonstrate the effectiveness of the proposed algorithm.

**Weaknesses:**

1) It is unclear how the 1D Gaussian kernel filter can be justified from the optimization perspective. A potential issue is that such a filter can be model-dependent and/or data-dependent, as shown in the ablation study that vision models benefit significantly more than the language model.
2) Theorems 2 and 3 seem superficial. They only present an ideal case; but they do not consider DP noise, the federated learning distributed nature, or the 1D Gaussian kernel filter.
3) The LLM-related experiments can be extended to more recent tasks with more recently published models.

**Questions:**

1) What is the intuition and theoretical justification for the 1D Gaussian kernel filter? Why can the adjacent element row/columns-wise be used to smoothing?
2) How can Theorems 2 and 3  explain the utility of the proposed method?
3) Does the proposed LA-LoRA method have different utility depending on the model and data?

---

> ### Author Response · Authors · 2025-11-25
> **Response (1/3)**
>
> We sincerely thank you for the insightful comments.
>
> **Response for Weakness 1:**
>
> Thank you for pointing this out.
>
> The 1D Gaussian filter is a way to impose a smoothness prior on the LoRA parameters. In LA-LoRA, we apply the filter along the input/output feature dimensions of the LoRA matrices (row-wise for $A$ and column-wise for $B$), so that smoothing acts within each feature/channel axis while keeping different low-rank components independent. Concretely, let $w$ be the parameters vectorized along this smoothed dimension and $g_i$ the corresponding noisy gradient. Using the filtered gradient $\widehat g_i = G_s * g_i$ can be interpreted as optimizing
> $
> \min_{w} \mathcal{L}(w) + \tau \sum_{i} (w_i - w_{i-1})^2,
> $
> i.e., a standard quadratic smoothness regularizer that penalizes differences between neighboring coordinates of $w$. The low-pass filter suppresses high-frequency perturbations introduced by DP noise and client heterogeneity while preserving the underlying structure of $A$ and $B$. We clarify this interpretation in Section 4.2.
>
> Regarding the model/data dependency, our experimental results show larger gains on vision models than on the language model. **This is mainly because the vision tasks and architectures are harder and more sensitive to DP noise**. For Swin-B on CIFAR-100 and Tiny-ImageNet, the high-dimensional vision representations and multi-class objectives cause DP training to severely degrade performance, leaving substantial room for improvement. In contrast, for RoBERTa on language benchmarks, the DP baselines are already relatively strong and close to each other under the same backbone, so there is less headroom for further gains. Nevertheless, LA-LoRA consistently matches or improves over all DP baselines across both vision and language models.
>
> > **Weakness 2.** Theorems 2 and 3 seem superficial. They only present an ideal case; but they do not consider DP noise, the federated learning distributed nature, or the 1D Gaussian kernel filter.
>
> **Response for Weakness 2:**
> We thank the reviewer for raising this point. We clarify the role of Theorems 2 and 3 as follows:
>
> Theorems 2 and 3 are intended as **an idealized structural analysis**, whose goal is to compare **alternating** versus **simultaneous** LoRA updates in terms of optimization geometry and stability, rather than to provide a full DPFL convergence theorem that simultaneously incorporates DP noise, federated distribution, and filtering.
>
> When DP is introduced, we simply replace the clean gradient by a ''true gradient + DP noise'' term:
>
> **In Theorem 2, the ''low-rank subspace projection'' still acts on this noisy gradient;**
>
> **In Theorem 3, the ''harmful cross term'' that is specific to joint updates still appears, while alternating updates still avoid this term, so the structural conclusions remain unchanged.**
>
> The DP mechanism and federated characteristics (clipping, noise addition, local client updates, and aggregation) are handled explicitly by Theorem 1, the algorithm description, and the experimental setup. Theorems 2 and 3 are abstractions for the single-client update; their conclusions can be directly extended to the federated setting via linear aggregation.
>
> **The 1D Gaussian kernel filter** is a linear **post-processing** step applied to already privatized gradients. It does not weaken the DP guarantees and is used in practice to suppress high-frequency noise; its effectiveness is demonstrated in our ablation studies.
>
> In the revised manuscript, we will state explicitly that:
>
> - Theorems 2 and 3 are idealized structural results focusing on the geometric and stability differences between alternating and simultaneous updates.
>
> - We add a remark after Theorem 3 explaining how the conclusions remain valid when the gradients are replaced by DP-noisy gradients and under federated aggregation.
>
> - We add a corollary that provides a comparative analysis with FFA-LoRA and RoLoRA.
>
> - We add a convergence Theorem.

---

> > ### Author Response · Authors · 2025-11-25
> > **Response (2/3)**
> >
> > Thank you for your comments.
> > > **Weakness 3.** The LLM-related experiments can be extended to more recent tasks with more recently published models.
> >
> > **Response for Weakness 3:**
> > In the revised version, we report DP federated fine-tuning results of LLaMA-2-7B on GLUE (**Appendix~B.4, Table 12**). We follow exactly the same fine-tuning hyperparameters and training pipeline as in our main experiments.
> >
> >
> > Table 12: Test accuracy (%) of Llama-2-7B on SST-2, QNLI, QQP and MNLI with $\epsilon=1$.
> >
> > | Model      | Method   | SST-2 | QNLI  | QQP   | MNLI  | Avg  |
> > |-----------|----------|-------|-------|-------|-------|------|
> > | Llama-2-7B| DP-LoRA  | 91.56 | 88.22 | 85.56 | 86.86 | 88.05 |
> > |           | FFA-LoRA | 92.53 | 89.23 | 85.56 | 86.98 | 88.58 |
> > |           | RoLoRA   | 92.12 | 89.34 | 85.98 | 87.21 | 88.66 |
> > |           | LA-LoRA  | 93.36 | 89.78 | 86.75 | 87.56 | 89.36 |
> >
> >
> > As shown in Table 12, LA-LoRA achieves the best average accuracy (89.36%), improving over DP-LoRA by 1.31\% and over FFA-LoRA and RoLoRA by 0.78% and 0.70%, respectively. These results demonstrate that LA-LoRA remains effective for modern LLMs.
> >
> >
> > > **Question 1.** What is the intuition and theoretical justification for the 1D Gaussian kernel filter? Why can the adjacent element row/columns-wise be used to smoothing?
> >
> > **Response for Question 1:**
> > Please see our response to **Weakness 1** for the optimization-based justification of the 1D Gaussian filter as a smoothness regularizer. Here we discuss Why can the adjacent element row/columns-wise be used to smoothing.
> >
> > In the LoRA decomposition $W = W\_0 + sBA$ for a linear map $\mathbb{R}^n \to \mathbb{R}^m$, the matrix $A \in \mathbb{R}^{r \times n}$ parameterizes low-rank directions over the **input feature dimension $n$**, while $B \in \mathbb{R}^{m \times r}$ parameterizes the same low-rank directions over the **output feature dimension $m$**. The rank dimension $r$ itself is just an index of basis vectors and does not carry a natural notion of locality. For this reason, we smooth $A$ **row-wise** (i.e., along its columns) so that the Gaussian filter acts along the input feature axis, and we smooth $B$ **column-wise** (i.e., along its rows) so that the filter acts along the output feature axis. In both cases, smoothing is applied within a meaningful feature/channel dimension, while different low-rank components (the $r$ dimension) are kept independent.

---

> > > ### Author Response · Authors · 2025-11-25
> > > **Response (3/3)**
> > >
> > > Thank you for your feedback.
> > > > **Question 2.** How can Theorems 2 and 3 explain the utility of the proposed method?
> > >
> > > **Response for Question 2:**
> > >
> > > **Theorem 2: Alternating low-rank updates = structured projection onto a task-relevant subspace (capacity control under noise).**
> > > Theorem 2 shows that alternating LoRA updates are equivalent to projecting the (possibly DP-noisy) gradient onto the low-rank subspace spanned by the LoRA factors. This has two direct implications for utility in DPFL:
> > >
> > > **(1) More efficient use of the limited ''privacy signal budget''.**
> > > Once DP noise is added, the signal-to-noise ratio of the gradient is already reduced. By constraining updates to a low-rank subspace that is learned and shaped by the task, LA-LoRA concentrates the limited update capacity on ''useful directions'' that recur across rounds and clients, instead of spreading noisy updates over the full high-dimensional parameter space.
> > >
> > > **(2) Implicit denoising effect.**
> > > Projecting the privatized gradient (true gradient + DP noise) onto a low-dimensional subspace effectively preserves the signal aligned with that subspace while partially cancelling isotropic noise. This leads to better downstream accuracy under the same $(\epsilon,\delta)$ privacy budget.
> > >
> > > **Theorem 3: Removing harmful cross terms = stability in multi-round DPFL training.**
> > > Theorem 3 shows that simultaneous low-rank updates introduce an additional cross term of the same order as the main update, while the alternating update in LA-LoRA completely removes this term. For DPFL utility, this implies:
> > >
> > > **(1) More stable feature learning.**
> > > The cross term drives features to drift toward directions that are not aligned with the task gradient. In a federated, multi-round setting with noisy local updates, this drift accumulates across rounds and amplifies the negative impact of DP noise.
> > >
> > > **(2) Mitigating the compound effect of ''noise × non-IID heterogeneity".**
> > > By eliminating this cross term, LA-LoRA's feature updates behave more like those of a well-conditioned full-parameter optimizer, avoiding unstable couplings between DP noise and client heterogeneity. This explains the substantial utility gains we observe in non-IID and data-scarce regimes.
> > >
> > > **Connection to experimental results.**
> > > Taken together, Theorems 2 and 3 predict that:
> > > - LA-LoRA should be more robust to DP noise;
> > > - Its advantage over baselines should become more pronounced when federated training runs longer and privacy constraints are stricter.
> > >
> > > This is exactly what we observe empirically: across multiple vision and language benchmarks, under the same $(\epsilon,\delta)$, LA-LoRA consistently achieves higher accuracy, and the gap to DP-LoRA and other baselines further widens when the privacy budget decreases or the number of training rounds increases.
> > >
> > >
> > > > **Question 3.** Does the proposed LA-LoRA method have different utility depending on the model and data?
> > >
> > > **Response for Question 3:**
> > > We agree that the utility gains of LA-LoRA differ across models and datasets, but this does not mean that the Gaussian filter is tied to a specific architecture or data domain. Rather, it reflects how strongly DP training affects each setting. Vision models in our DPFL setup are much more severely degraded by DP noise, so there is substantial room for improvement. In contrast, the language model starts from a much stronger pre-trained backbone and remains relatively robust under DP, which naturally limits the achievable gains. Importantly, across all tested architectures, whether in DP or non-private settings, LA-LoRA consistently matched or outperformed the strongest baseline method.

---

> > > > ### Author Response · Authors · 2025-11-28
> > > > **Thank you very much！Reviewer gf7V**
> > > >
> > > > We are grateful for your comments and the time you spent on our work. We hope our replies have solved your concerns.

---

### Official Review · Reviewer_kNbJ · 2025-10-29

**Soundness:** 3
**Presentation:** 3
**Contribution:** 3
**Rating:** 4
**Confidence:** 3

**Summary:**

This paper investigates the limitations of directly applying Low-Rank Adaptation (LoRA) in differentially private federated learning (DPFL). The authors identify three intrinsic challenges that arise when LoRA is used under DP constraints: (1) gradient coupling between asymmetric matrices, (2) structural amplification of DP noise due to multiplicative interactions between noise terms, and (3) sharpness in global aggregation that leads to unstable convergence and poor generalization.

To mitigate these issues, they propose LA-LoRA, a simple yet effective modification that alternates updates between LoRA’s two low-rank matrices within each local round and applies an optional Gaussian low-pass filter to smooth noisy gradients. Theoretical analysis establishes convergence stability and privacy guarantees. Experiments on both vision and language models show consistent improvements across multiple privacy budgets, achieving state-of-the-art accuracy while maintaining formal DP guarantees.

Overall, the paper is well-written, methodologically sound, and provides both theoretical and empirical support. Its main contribution lies in identifying the underexplored failure modes of LoRA under DPFL and offering an elegant yet practical fix.

**Strengths:**

1. **Clear problem identification.**
The authors provide an intuitive and thorough analysis of why existing DP-LoRA methods fail, including gradient coupling, multiplicative noise amplification, and curvature-related instability after aggregation — each well-motivated and demonstrated through figures and equations.

2. **Sound theoretical grounding.**
The paper presents convergence and privacy guarantees with clear mathematical derivations and ties them directly to the algorithmic design. This theoretical foundation strengthens the credibility of the proposed solution.

3. **Comprehensive experiments.**
The work evaluates both language and vision tasks using multiple baselines under varying ε-budgets. The improvements are consistent and substantial.

4. **Simple yet effective modification.**
The proposed alternating update and Gaussian filter are computationally lightweight, easy to integrate, and yield significant utility gains without sacrificing privacy guarantees.

**Weaknesses:**

1. Based on your deduction, the identified issues (gradient coupling, DP-noise amplification, sharpness) seem to arise from LoRA’s intrinsic structure rather than federated communication. Do these problems persist even in centralized DP-LoRA? The paper should clarify that. If so, the research should focus on centralized DP-LoRA, instead of the federated setting.

2. While LA-LoRA achieves higher cosine similarity, it is unclear whether this causally leads to better accuracy or is merely correlated. Including a non-DP Fed-LoRA baseline could help isolate whether high similarity is inherent and thus beneficial.

3. While the low-pass filter idea is interesting, the paper should better differentiate it from recent filtered gradient mechanisms such as [1], which also employ noise smoothing for DP stability.

4. Although LA-LoRA is efficient, it adds alternation and filtering. A detailed breakdown of time/memory cost versus baselines (beyond Table 6) would strengthen the practical claim.

**Reference**

[1] *Doppler: Differentially private optimizers with low-pass filter for privacy noise reduction. Neurips 2024*

**Questions:**

1. Are the three limitations (gradient coupling, noise amplification, sharpness) unique to the federated setup, or do they also appear in centralized DP-LoRA? If they appear in both, could LA-LoRA also enhance centralized DP-LoRA performance?

2. Figure 2 compares DP-LoRA and LA-LoRA, but not Fed-LoRA (non-DP). Does higher gradient-similarity appear in the non-DP case? This would clarify whether cosine alignment is indeed the key to improved performance.

3. What is the theoretical justification for the specific kernel? Have the authors considered learned or adaptive filters, or tested sensitivity to the kernel width?

I would raise my score if the above questions are addressed properly.

---

> ### Author Response · Authors · 2025-11-25
> **Response (1/3)**
>
> We sincerely thank you for the insightful comments.
>
> > **Weakness 1.** Based on your deduction, the identified issues (gradient coupling, DP-noise amplification, sharpness) seem to arise from LoRA’s intrinsic structure rather than federated communication. Do these problems persist even in centralized DP-LoRA? The paper should clarify that. If so, the research should focus on centralized DP-LoRA, instead of the federated setting.
>
> **Response for Weakness 1:**
> We agree that the identified issues come from the structure of DP-LoRA. They are not only caused by federated communication. While prior work has studied DP fine-tuning with LoRA in centralized settings, it mainly treats LoRA as a convenient parametrization and reports end-to-end accuracy. To the best of our knowledge, neither centralized DP-LoRA nor federated DP-LoRA approaches have analyzed gradient coupling and sharp minima induced by low-rank updates.
>
> We therefore focus on the DP federated learning (DPFL) setting for two reasons. **First**, DPFL is both practically important and substantially more challenging: non-iid client data and DP noise amplify these structural issues and make them much more harmful in practice. **Second**, one of our key observations is that sharpness arising from aggregating heterogeneous low-rank factors is inherently tied to federated aggregation. We state this in **Section 3** by explicitly stating that DP interacts with the low-rank parameterization and that these failure modes are further amplified in federated learning due to data heterogeneity and noisy local updates. In addition, we include centralized DP-LoRA results in **Appendix A.6**.
>
> In Appendix A.6, we add centralized DP experiments with the same architecture, optimizer, clipping norm, and noise multiplier as in the federated setup. In the centralized case, LA-LoRA(-filter) improves test accuracy by 1.18\% over DP-LoRA. In the federated case, LA-LoRA(-filter) improves test accuracy by 11.22\% over DP-LoRA, and the late gradient cosine also increases (ΔCos = 0.108). These results show that the DP–LoRA issues already exist in centralized training. They become much more severe in the federated setting, where LA-LoRA(-filter) brings larger benefits.
>
> **Table 9: Centralized and federated DP training for Swin-T on CIFAR-100 with $\epsilon = 3$.
> ''Grad. Cos. (late)'' denotes the average cosine similarity between $\nabla_A \mathcal{L}$ and $\nabla_B \mathcal{L}$ over the last 10\% of training steps.**
>
> | Setting      | Method            | Test Acc. (\%)     | $\Delta$Acc     | Grad. Cos. (late) | $\Delta$Cos     |
> |-------------|-------------------|--------------------|-----------------|---------------------|-----------------|
> | Centralized | DP-LoRA           | $76.11 \pm 0.38$               | –               | $0.681$               | –               |
> |  | LA-LoRA(-filter)  | $77.29 \pm 0.54$   | $\uparrow 1.18$ | $0.713 $              | $\uparrow 0.032$|
> | Federated   | DP-LoRA           | $45.40 \pm 0.40$   | –               | $0.337 $              | –               |
> |   | LA-LoRA(-filter)  | $56.62 \pm 0.54$   | $\uparrow 11.22$| $0.445$               | $\uparrow 0.108$|
>
>
> > **Weakness 2.** While LA-LoRA achieves higher cosine similarity, it is unclear whether this causally leads to better accuracy or is merely correlated. Including a non-DP Fed-LoRA baseline could help isolate whether high similarity is inherent and thus beneficial.
>
> **Response for Weakness 2:**
> To clarify this, we add **non-private federated experiments** in **Appendix A.7 (Table 10)**.
>
> **Table 10: Non-private federated training for Swin-T on CIFAR-100.
> “Grad. Cos. (late)” denotes the average cosine similarity between $\nabla_A\mathcal{L}$ and $\nabla_B\mathcal{L}$ over the last 10\% of training steps.**
>
> | Method           | Test Acc. (\%)  | $\Delta$Acc      | Grad. Cos. (late) | $\Delta$Cos      |
> |------------------|------------------|------------------|---------------------|------------------|
> | Fed-LoRA         | $90.56 \pm 0.22$      | -                | $0.694$                | -                |
> | LA-LoRA(-filter) | $91.25 \pm 0.15$      | $\uparrow 0.69$   | $0.783$                | $\uparrow 0.089$   |
>
> In these experiments, we remove DP noise and clipping (no DP, $\sigma = 0$). LA-LoRA(-filter) still achieves higher test accuracy and higher late gradient cosine than Fed-LoRA. This shows that gradient alignment is beneficial even in the non-private setting and is not only a side effect of DP noise. DP further degrades alignment, but the underlying relation between alignment and performance already exists without DP.

---

> ### Author Response · Authors · 2025-11-25
> **Response (2/3)**
>
> We greatly appreciate your careful review and detailed comments.
> > **Weakness 3.** While the low-pass filter idea is interesting, the paper should better differentiate it from recent filtered gradient mechanisms such as [1], which also employ noise smoothing for DP stability.
>
> **Response for Weakness 3:**
> We thank the reviewer for pointing out the relation to Doppler [1]. Doppler smooths the full gradients at the optimizer level. Our filter instead smooths the LoRA updates in the low-rank subspace and is coupled with the alternating update of $A$ and $B$. Thus it is a different mechanism, but can be combined with Doppler. We add an ablation on smoothing strategies in **Appendix C.2 (Table 15)**. We compare three options in the same DP-FL setting: no smoothing, Doppler, and our Gaussian low-pass filter, for both DP-LoRA and LA-LoRA.  For more details, please see Appendix C.2.
>
> **Table 15: Impact of different low-pass filters on federated LoRA performance (\%) across GLUE and image classification benchmarks, $\epsilon = 3$.**
>
> | Method        | QQP (RoBERTa-Base)| MNLI (RoBERTa-Base)  | CIFAR-100 (Swin-B) | Tiny-ImageNet (Swin-B) |
> |--------|---------|---------|--------|-----------|
> | DP-LoRA               | $84.56\pm 0.83$   | $80.98\pm 0.44$     | $56.52\pm 0.51$       | $30.64\pm 0.30$      |
> | DP-LoRA(+filter)     | $84.79\pm 0.42$  | $81.43\pm 0.57$     | $69.08\pm 0.52$     | $49.85\pm 0.55$     |
> | DP-LoRA(+Doppler)    | $84.82\pm 0.49$ | $81.04\pm 0.57$    | $66.12\pm 0.71$    | $48.53\pm 0.57$    |
> | LA-LoRA(-filter)       | $84.98\pm 0.51$    | $82.40\pm 0.53$     | $70.38\pm 0.48$         | $53.07\pm 0.60$           |
> | **LA-LoRA**       | $\mathbf{85.83}{\pm 0.49}$ | $\mathbf{82.99}{\pm 0.42}$ | $\mathbf{75.29}{\pm 0.35}$ | $\mathbf{61.97}{\pm 0.56}$ |
> | LA-LoRA(+Doppler)     | $85.45\pm 0.50$     | $82.13\pm 0.58$     | $72.88\pm 0.56$    | $57.36\pm 0.66$  |
>
> On GLUE with RoBERTa-Base, both Doppler and our filter improve DP-LoRA, and our filter is slightly better. On the harder Swin-B vision tasks, our filter brings much larger gains. It raises DP-LoRA from 56.52\% to 69.08\% on CIFAR-100 and from 30.64\% to 49.85\% on Tiny-ImageNet, while Doppler reaches 66.12\% and 48.53\%.  With local alternating updates, LA-LoRA with our filter achieves the best results (75.29\% on CIFAR-100 and 61.97\% on Tiny-ImageNet), outperforming LA-LoRA(+Doppler) by **2.41\%** and **4.61\%**.
>
>
> > **Weakness 4.** Although LA-LoRA is efficient, it adds alternation and filtering. A detailed breakdown of time/memory cost versus baselines (beyond Table 6) would strengthen the practical claim.
>
> **Response for Weakness 4:**
> In the revision, we update Table 7 in Appendix A.4.
>
> **Table7: Per-round computation cost, memory cost and performance comparison of Swin-B at $\epsilon = 1$.**
>
> | Method            | Time Cost (s) CIFAR-100 | Time Cost (s) Tiny-ImageNet | Memory Cost (MB) CIFAR-100 | Memory Cost (MB) Tiny-ImageNet | Test Accuracy (%) CIFAR-100 | Test Accuracy (%) Tiny-ImageNet |
> |-------------------|---------------------|----------------------|-----------------------|-----------------------|-----------------------|----------------------------|
> | DP-LoRA           | 30.35                   | 28.02                       | 3524                        | 3524                           | 55.98                        | 30.20                           |
> | DP-LoRA(+filter)  | 30.72                   | 28.51                       | 3524                        | 3524                           | 67.95                        | 48.09                           |
> | FFA-LoRA          | 17.85                   | 16.54                       | 1762                        | 1762                           | 61.94                        | 39.33                           |
> | RoLoRA            | 16.64                   | 16.32                       | 1762                        | 1762                           | 67.88                        | 43.85                           |
> | LA-LoRA(-filter)  | 17.30                   | 17.16                       | 1762                        | 1762                           | 69.87                        | 52.72                           |
> | **LA-LoRA**       | 17.44                   | 17.23                       | 1762                        | 1762                           | 74.56                        | 60.68                           |
>
>
> Compared to standard DP-LoRA, LA-LoRA reduces the per-round time from 30.35 s to 17.44 s on CIFAR-100 and from 28.02 s to 17.23 s on Tiny-ImageNet, and halves the memory footprint from 3524 MB to 1762 MB due to the low-rank parametrization. Among low-rank DP baselines (FFA-LoRA, RoLoRA, and LA-LoRA(-filter)), LA-LoRA has per-round time that is comparable to the fastest method, differing by at most about 1 s, while delivering the highest accuracy on both datasets. The additional Gaussian smoothing filter incurs only a negligible overhead.

---

> ### Author Response · Authors · 2025-11-25
> **Response (3/3)**
>
> Thank you for your comments.
> > **Question 1.** Are the three limitations (gradient coupling, noise amplification, sharpness) unique to the federated setup, or do they also appear in centralized DP-LoRA? If they appear in both, could LA-LoRA also enhance centralized DP-LoRA performance?
>
>
>
> **Response for Question 1:**
> This question is closely related to **Weakness 1**. As we clarify there and in **Appendix A.6**, these limitations also appear in centralized DP-LoRA. We added centralized DP experiments and show that LA-LoRA(-filter) improves centralized DP-LoRA as well, though the gains are smaller than in DPFL.
>
>
> > **Question 2.** Figure 2 compares DP-LoRA and LA-LoRA, but not Fed-LoRA (non-DP). Does higher gradient-similarity appear in the non-DP case? This would clarify whether cosine alignment is indeed the key to improved performance.
>
> **Response for Question 2:**
> Please see our response to **Weakness 2** and **Appendix A.7**. We add non-private federated experiments and show that LA-LoRA(-filter) also achieves higher gradient cosine similarity and accuracy than Fed-LoRA without DP.
>
> > **Question 3.** What is the theoretical justification for the specific kernel? Have the authors considered learned or adaptive filters, or tested sensitivity to the kernel width?
>
> **Response for Question 3:** In LA-LoRA, we use the kernel $G\_s = \tfrac{1}{16}[1,4,6,4,1]$. This is the standard 5-tap binomial filter, which is a discrete approximation of a 1D Gaussian kernel and is widely used as a simple low-pass filter. The DP mechanism adds independent Gaussian noise to LoRA gradients, while neighboring entries in $A$ and $B$ are typically similar. Under these conditions, smoothing with a Gaussian kernel is a natural and well-motivated way to attenuate high-frequency DP noise while preserving the main structure of the gradients.
>
> We agree that learned or adaptive filters are an interesting direction. In this work, our main goal is to keep LA-LoRA simple and lightweight for DP federated training, so we use a fixed kernel to avoid additional trainable parameters and hyperparameter tuning. We plan to explore adaptive or learned kernels in future work.
>
> We also conduct a sensitivity study on different kernel widths, and report the results in **Appendix D.2 (Table 19)**. Besides our default 5-tap kernel, we further evaluate a narrower 3-tap kernel and a wider 7-tap kernel on CIFAR-100 and Tiny-ImageNet. As shown in Table 19, the performance of LA-LoRA remains stable across these configurations: the test accuracy varies within at most **0.97%**. Even with the worst-performing 3-tap kernel, LA-LoRA improves over the strongest baseline RoLoRA (67.88\% on CIFAR-100 and 43.85\% on Tiny-ImageNet) by 5.66\% and 16.07\%, respectively.
>
> **Table 19: Sensitivity of LA-LoRA to the choice of 1D binomial kernel size. Results are reported for Swin-B on CIFAR-100 and Tiny-ImageNet at $\epsilon = 1$.**
>
> | Kernel | Size | Coefficients                                   | CIFAR-100  | Tiny-ImageNet  |
> |-----------------|--------------|----------------------------------------------------------|------------------------------|----------------------------------|
> | $G_s^{(3)}$       | 3            | $\tfrac{1}{4}[1, 2, 1]$                                  | $73.59{\pm 0.60}$           | $59.92{\pm 0.51}$               |
> | $G_s^{(5)}$       | 5            | $\tfrac{1}{16}[1, 4, 6, 4, 1]$                           | $\mathbf{74.56}{\pm 0.52}$  | $60.68{\pm 0.55}$               |
> | $G_s^{(7)}$       | 7            | $\tfrac{1}{64}[1, 6, 15, 20, 15, 6, 1]$                  | $73.88{\pm 0.46}$           | $\mathbf{60.74}{\pm 0.63}$      |

---

> > ### Author Response · Authors · 2025-11-28
> > **Thank you very much！Reviewer kNbJ**
> >
> > We are grateful for the time and effort you devoted to evaluating our manuscript. We hope that our clarifications and revisions satisfactorily respond to your comments.

---

### Official Review · Reviewer_fjPf · 2025-10-29

**Soundness:** 3
**Presentation:** 4
**Contribution:** 2
**Rating:** 4
**Confidence:** 4

**Summary:**

This paper revisits the role of LoRA within the context of privacy-aware FL. The method introduces a local alternating update scheme where the two LoRA matrices are updated in turn during local training to decouple gradients and reduce noise amplification. Additionally, a low-pass Gaussian smoothing filter is applied to suppress high-frequency noise before aggregation, improving stability and generalization.

**Strengths:**

1. Strong Empirical Performance: Extensive experiments demonstrates high accuracy while achieving better privacy–utility balance than baseline defenses.

2. Incorporation of Low-Pass Smoothing Filter: Uses a Gaussian low-pass filter before aggregation to smooth high-frequency DP noise, improving stability of the algorithm.

**Weaknesses:**

1. Lack of studies on different local alternating strategies: The current method fixes the local alternating update pattern to update matrix A on odd iterations and matrix B on even iterations, effectively using a one-step inner loop. It would strengthen the work to explore and experiment with other strategies such as updating A for several consecutive steps before switching to update B, or vice versa. Moreover, analysis of how different alternating schedules affect performance under various degrees of data heterogeneity would provide deeper insights.

2. Incomplete Theoretical Convergence Analysis: Theorems 2 and 3 provide structural insights into projected gradients and stability but fall short of delivering explicit convergence rates or optimization behavior under privacy noise. Deriving explicit convergence rates, possibly for both convex and non-convex settings, and clarifying the trade-off between differential privacy noise injection and optimization accuracy, would provide a more comprehensive understanding of algorithm performance.

3. Unclear Privacy Guarantees Regarding the Smoothing Filter: The privacy guarantee in Theorem 1 appears to ignore the effect of the low-pass Gaussian smoothing filter applied to the gradients before aggregation. It raises the question of whether the differential privacy guarantee strictly holds when the smoothing filter is included. Since the filter alters the noise characteristics, a clear discussion or proof of how these impacts or preserves privacy guarantees would strengthen the claim. Without this, the smoothing filter may be viewed as a redundant or questionable design that could be alternatively addressed by carefully tuning the privacy budgets.

**Questions:**

1. The word "update" is spelled incorrectly in the beginning of section 4.1.

**Details Of Ethics Concerns:**

No concerns.

---

> ### Author Response · Authors · 2025-11-25
> **Response (1/2)**
>
> We thank you for the helpful comments and suggestions.
>
> > **Weakness 1.** Lack of studies on different local alternating strategies: The current method fixes the local alternating update pattern to update matrix A on odd iterations and matrix B on even iterations, effectively using a one-step inner loop. It would strengthen the work to explore and experiment with other strategies such as updating A for several consecutive steps before switching to update B, or vice versa. Moreover, analysis of how different alternating schedules affect performance under various degrees of data heterogeneity would provide deeper insights.
>
> **Response for Weakness 1:**
> In Section 6.3, we add a study of different local alternating update strategies between the two LoRA factors under a fixed privacy budget ($\epsilon = 3$) and the same total number of local steps. Specifically, we compare four strategies: **1-step $B$ / 1-step $A$** (our original/default odd–even scheme), **2-step $B$ / 2-step $A$**, **5-step $B$ / 5-step $A$**, and the reversed variant **5-step $A$ / 5-step $B$**, on CIFAR-100 and Tiny-ImageNet with Dirichlet $\beta=0.6$ and $\beta=0.1$ (see Table~6).
>
> **Table 6: Effect of different local alternating strategies on CIFAR-100 and Tiny-ImageNet (Swin-B) under $\epsilon=3$ and different Dirichlet distributions. Strategy “k-step B / k-step A” denotes performing k local gradient steps on B followed by k steps on A in each local round.**
>
> | Strategy                        | Dirichlet $\beta=0.6$ CIFAR-100 | Dirichlet $\beta=0.6$ Tiny-ImageNet | Dirichlet $\beta=0.1$ CIFAR-100 | Dirichlet $\beta=0.1$ Tiny-ImageNet |
> |-------------------------|-----------------------------|--------------------|-----------------|-----------------------------|
> | 1-step B / 1-step A (default)   | **89.62±0.37**              | **80.77±0.50**                   | 75.29±0.35                  | **61.97±0.56**                    |
> | 2-step B / 2-step A             | 89.29±0.47                  | 80.58±0.39                       | 75.15±0.38                  | 61.36±0.42                        |
> | 5-step B / 5-step A             | 89.31±0.42                  | 80.72±0.40                       | **75.47±0.46**              | 61.78±0.51                        |
> | 5-step A / 5-step B             | 89.45±0.26                  | 80.70±0.42                       | 75.34±0.44                  | 61.79±0.58                        |
>
> Across all four settings, the accuracy gap between the best and worst schedule is always below 0.7\% and typically within 0.3\%. Our default **1-step $B$ / 1-step $A$** strategy achieves the best performance in most cases and is at most 0.2\% worse than the best variant otherwise. These results show that LA-LoRA is not sensitive to the precise local alternating pattern and that its performance gains mainly stem from alternating optimization of the two LoRA factors, rather than from a carefully tuned update strategy, even under different levels of data heterogeneity.
>
> > **Weakness 2.** Incomplete Theoretical Convergence Analysis: Theorems 2 and 3 provide structural insights into projected gradients and stability but fall short of delivering explicit convergence rates or optimization behavior under privacy noise. Deriving explicit convergence rates, possibly for both convex and non-convex settings, and clarifying the trade-off between differential privacy noise injection and optimization accuracy, would provide a more comprehensive understanding of algorithm performance.
>
> **Response for Weakness 2:**
> We thank the reviewer for the helpful comment. Beyond the structural stability results in Theorems 2 and 3, our paper already provides an explicit convergence guarantee for LA-LoRA in **Theorem 4 (Appendix E.4)**. Under the rank-$r$ RIP assumption (Appendix E.4.2) and a suitable stepsize $\eta$, noiseless LA-LoRA without momentum satisfies linear convergence: $\mathcal{L}\_c(B_{k+1},A\_{k+1}) \le (1-\eta\_c)^2 \\mathcal{L}\_c(B\_k,A\_k),$
> and the reconstruction error decays exponentially:
> $\Bigl\\|\textstyle\sum\_{i=1}^P B\_k^i A\_k^i - X\_\star\Bigr\\|\_F^2 \le \frac{1+\delta\_r}{1-\delta\_r} (1-\eta\_c)^{2k} \Bigl\\|\textstyle\sum\_{i=1}^P B\_0^i A\_0^i - X\_\star\Bigr\\|\_F^2,
> $ where $\eta\_c = 2P(1-\delta\_r)\Bigl(\eta - \frac{\eta^2(1+\delta\_r+\tfrac{1}{P})}{2}\Bigr).$ In the revision, we will move this result closer to the main text and clearly state that it characterizes the noiseless optimization behavior of LA-LoRA.
>
> For the DP-noisy case, we will add a corollary based on Theorem 4: if each step adds zero-mean DP noise with variance $\sigma^2$, then $\mathbb{E}[\mathcal{L}\_c(B\_{k+1},A\_{k+1})] \le (1-\eta\_c)^2 \\mathbb{E}[\mathcal{L}\_c(B\_k,A\_k)] + C\sigma^2,$
> for a constant $C$ depending on problem parameters. Hence LA-LoRA converges linearly to an $O(\sigma^2)$ neighborhood of the optimum, making the privacy–utility trade-off explicit.

---

> ### Author Response · Authors · 2025-11-25
> **Response (2/2)**
>
> Thank you for your detailed feedback.
> > **Weakness 3.** Unclear Privacy Guarantees Regarding the Smoothing Filter: The privacy guarantee in Theorem 1 appears to ignore the effect of the low-pass Gaussian smoothing filter applied to the gradients before aggregation. It raises the question of whether the differential privacy guarantee strictly holds when the smoothing filter is included. Since the filter alters the noise characteristics, a clear discussion or proof of how these impacts or preserves privacy guarantees would strengthen the claim. Without this, the smoothing filter may be viewed as a redundant or questionable design that could be alternatively addressed by carefully tuning the privacy budgets.
>
> **Response for Weakness 3:**
> We clarify this issue in **Section 5** and **Appendix E.1** of the revised manuscript. In LA-LoRA, sample-level Gaussian noise is first injected into the clipped client updates. This noisy mechanism is exactly what is analyzed in Theorem 1. The subsequent Gaussian low-pass smoothing filter is applied after noise addition as a deterministic linear transform that depends only on public hyperparameters and is independent of the private data. By the standard **post-processing invariance** of differential privacy, any such (possibly randomized) mapping $f$ composed with a DP mechanism $\mathcal{M}$, i.e., $f \\circ\\mathcal{M}$, preserves the same $(\epsilon,\delta)$ (or RDP) guarantee. Therefore, the Gaussian smoothing filter does not weaken nor alter the privacy guarantees stated in Theorem 1; it only serves as a post-processing step that shapes the spectrum of the injected noise and can improve utility without consuming additional privacy budget.
>
>
>
> > **Question 1.** The word "update" is spelled incorrectly in the beginning of section 4.1.
>
> **Response for Question 1:** Thank you for pointing out this typo. We have corrected the spelling of the word "update'' at the beginning of Section 4.1 in the revised manuscript.

---

> > ### Author Response · Authors · 2025-11-28
> > **Thank you very much！Reviewer fjPf**
> >
> > Thank you very much for your time and constructive comments. We hope that our responses adequately address your concerns.

---

### Author Response · Authors · 2025-12-01
**Summary of Novelty and Key Contributions**

Dear AC and Reviewers,

We sincerely thank you for the time and effort you devoted to reviewing our work. We truly appreciate the constructive and insightful comments, which have helped us substantially improve the quality and clarity of the paper. We briefly summarize the main novelty and contributions of our work.

---

1. **Rethinking LoRA under privacy-preserving federated learning.**
- We take DP-LoRA as our starting point and observe that it often suffers from unstable training and degraded performance under small privacy budgets and heterogeneous clients.
- We attribute this to three structural factors: gradient coupling, noise amplification, and increased global sharpness, and we find that federated heterogeneity further amplifies these issues.
- This motivates rethinking LoRA for privacy-preserving federated learning in large models.

2. **LA-LoRA: a framework for privacy-preserving federated LoRA.**
 - **Local alternating update strategy.**
     Within each local training round, only one LoRA factor is updated at a time while the other is kept fixed. This reduces gradient coupling, controls the propagation of DP noise, and alleviates sharp global aggregation.
- **Local Gaussian low-pass filtering before aggregation.**
     Each client optionally applies a 1D Gaussian low-pass filter to its privatized LoRA gradients before uploading them. This attenuates high-frequency perturbations introduced by DP noise, stabilizes aggregation, and preserves DP guarantees as a post-processing step.

3. **Theory and empirical validation.**
- **Theory.** We provide four main theorems and one corollary: a privacy theorem for the full pipeline (Theorem 1); a projected-gradient view with closed-form updates for the alternating scheme (Theorem 2); a feature-learning analysis showing the destabilizing effect of the cross term in simultaneous updates (Theorem 3), a convergence theorem showing geometric decay of the low-rank reconstruction error under suitable assumptions (Theorem 4), and a corollary comparing LA-LoRA with FFA-LoRA and RoLoRA.

- **Main empirical results.** We evaluate LA-LoRA under DPFL, centralized DP training, and non-private settings, on both vision and language tasks, and compare against strong baselines such as DP-LoRA, FFA-LoRA, and RoLoRA.

- Additional ablations on local update strategies, the use of Gaussian filtering, kernel width, and the number of local steps show that the two components of LA-LoRA are effective and robust under DPFL, and that the empirical trends are consistent with our theoretical insights.

Sincerely,
The Authors

---

> ### Author Response · Authors · 2025-12-01
> **Summary of Revisions in Response to Reviewers**
>
> Dear AC and Reviewers,
>
> In this part, we summarize the concrete revisions we have made in response to the reviewers’ comments.
>
> ---
> **1. Regarding Experimental Analysis**
>
> - In response to Reviewer **fjPf**’s **Weakness 1**, we added an ablation study on different local update strategies in **Section 6.3 (Table 6)**.
>
> - In response to Reviewer **kNbJ**’s **Weakness 1** and **Question 1**, we included centralized DP experiments in **Appendix A.6 (Table 9)** and clarified this at the beginning of **Section 3** in the main text.
>
> - In response to Reviewer **kNbJ**’s **Weakness 2** and **Question 2**, and Reviewer **cWrF**’s **Question 3**, we added non-private federated experiments for vision tasks in **Appendix A.7 (Tables 10 and 11)** and non-private federated experiments for language tasks in **Appendix B.5 (Table 13)**.
>
> - In response to Reviewer **kNbJ**’s **Weakness 3**, we added a comparison with the **Doppler** in **Appendix C.2 (Table 15)**.
>
> - In response to Reviewer **kNbJ**’s **Weakness 4**, we revised **Appendix A.4 (Table 7)** to provide a clearer comparison of training time and memory usage.
>
> - In response to Reviewer **kNbJ**’s **Question 3**, we conducted a sensitivity study on different Gaussian kernel widths and reported the results in **Appendix D.2 (Table 19)**.
>
> - In response to Reviewer **gf7V**’s **Weakness 3**, we added additional experiments using the Llama-2-7B model in **Appendix B.4 (Table 12)**.
>
> - In response to Reviewer **cWrF**’s **Weakness 2**, we added an analysis of the three identified challenges in **Appendix C.3 (Table 16)**.
>
> - In response to Reviewer **cWrF**’s **Question 4**, we included a sensitivity analysis of the number of local steps \(K\) in **Appendix C.4 (Table 17)**.
>
> ---
>
> **2. Regarding Theoretical Analysis**
>
> - In response to Reviewer **fjPf**’s **Weakness 2**, we added **Theorem 4** in **Section 5**, with a detailed proof provided in **Appendix E.4**.
>
> - In response to Reviewer **gf7V**’s **Weakness 2** and **Question 2**, and Reviewer **cWrF**’s **Weakness 1**, we revised the description in **Section 5**, added **Theorem 4**, introduced a new corollary in **Appendix E.5**, and provided additional explanations of Theorem 4 in **Appendix E.4**.
>
> ---
>
> **3. Regarding the Gaussian Low-Pass Filter**
>
> - In response to Reviewer **fjPf**’s **Weakness 3**, we clarified the influence of the Gaussian low-pass smoothing filter on the differential privacy guarantees in the discussion following **Theorem 1 in Section 5**, and provided additional details in **Appendix E.1**.
>
> - In response to Reviewer **gf7V**’s **Weakness 1** and **Question 1**, we revised and expanded the description of the filter in **Section 4.2**.
>
> ---
>
> We greatly appreciate every comment from the reviewers and are thankful for the suggestions that helped us improve the paper. We hope that these revisions address the main concerns and clarify our contributions.
>
> Sincerely,
> The Authors

---

### Meta-Review · Area_Chair_maPg · 2026-01-08

**Summary:**

The reviewers raised several key concerns that informed the initial borderline scores (two Weak Reject [4] and two Weak Accept [6]). The primary concerns centered on: 1) Empirical Validation: Several reviewers requested additional experiments to strengthen the claims, including comparisons in centralized DP and non-private federated settings, ablation studies on core components. 2) Reviewers noted a need for deeper theoretical grounding, specifically requesting a formal convergence analysis. 3) Concerns were raised regarding the novelty and impact of the Gaussian low-pass filter, its effect on Differential Privacy (DP) guarantees.

**Reviewer Concerns:**

Addressed Concerns:

-- Reviewer fjPf (Weakness 1, 2, 3): The authors added the requested ablation on local update strategies (Table 6), provided a new convergence theorem (Theorem 4) with proof, and clarified the DP-compatibility of the Gaussian filter as a post-processing step.

-- Reviewer kNbJ (Weakness 1, 2, 3, 4; Questions 1, 2, 3): The authors successfully added experiments for centralized DP (Appendix A.6), non-private FL for vision/language tasks (Appendix A.7, B.5), a comparison with Doppler (Appendix C.2), a clearer analysis of time/memory (Appendix A.4), and a sensitivity study on Gaussian kernel width (Appendix D.2).

-- Reviewer gf7V (Weakness 1, 2, 3; Questions 1, 2): The authors revised the description of the Gaussian filter, expanded the theoretical section with Theorem 4.

-- Reviewer cWrF (Weakness 1, 2; Questions 3, 4): The authors enhanced the theoretical analysis (Theorem 4, corollary), added non-private FL experiments, provided an explicit analysis of the three challenges (Appendix C.3), and included a sensitivity analysis on the number of local steps (Appendix C.4).

Outstanding Concerns:

Reviewer gf7V's point on filter novelty/impact: While the authors clarified the filter's role and DP properties, the reviewer's broader question about its fundamental novelty compared to generic smoothing techniques may remain a matter of perspective. However, its integration and demonstrated efficacy within the proposed LA-LoRA framework for DPFL are now well-supported.

**Reviewer Scores:**

Reviewer fjPf (Initial: 4 - Weak Reject): Likely to increase to a 6 (Borderline Accept).

Reviewer kNbJ (Initial: 4 - Weak Reject): Likely to increase to a 6 (Borderline Accept).

Reviewer gf7V (Initial: 6 - Weak Accept): Likely to remain at a 6 (Borderline Accept).

Reviewer cWrF (Initial: 6 - Weak Accept): Likely to remain at a 6 (Borderline Accept).

---

### Decision · Program_Chairs · 2026-01-26

Accept (Poster)